# Mechanisms of chromosome biorientation and bipolar spindle assembly analyzed by computational modeling

Christopher Edelmaier[1†], Adam R Lamson[1], Zachary R Gergely[1,2], Saad Ansari[1], Robert Blackwell[1], J Richard McIntosh[2], Matthew A Glaser[1], Meredith D Betterton[1,2]*

[1]Department of Physics, University of Colorado Boulder, Boulder, United States; [2]Department of Molecular, Cellular, and Developmental Biology, University of Colorado Boulder, Boulder, United States

**Abstract** The essential functions required for mitotic spindle assembly and chromosome biorientation and segregation are not fully understood, despite extensive study. To illuminate the combinations of ingredients most important to align and segregate chromosomes and simultaneously assemble a bipolar spindle, we developed a computational model of fission-yeast mitosis. Robust chromosome biorientation requires progressive restriction of attachment geometry, destabilization of misaligned attachments, and attachment force dependence. Large spindle length fluctuations can occur when the kinetochore-microtubule attachment lifetime is long. The primary spindle force generators are kinesin-5 motors and crosslinkers in early mitosis, while interkinetochore stretch becomes important after biorientation. The same mechanisms that contribute to persistent biorientation lead to segregation of chromosomes to the poles after anaphase onset. This model therefore provides a framework to interrogate key requirements for robust chromosome biorientation, spindle length regulation, and force generation in the spindle.

*For correspondence:
mdb@colorado.edu

Present address: [†]Department of Cell and Developmental Biology, University of Colorado Anschutz Medical Campus, Aurora, United States

**Competing interests:** The authors declare that no competing interests exist.

## Introduction

Cell biology seeks to understand how nanometer-scale molecules organize micron-scale cells, a question well-suited to theory and modeling (*Marshall, 2017*). As quantitative cell biology has grown, modeling has expanded in scope (*Mogilner et al., 2006*). Theory and simulation can now predict cellular phenomena across length and time scales, giving new insight into cellular self-organization. In the cytoskeleton, an important challenge is understanding how a relatively small number of building blocks can produce diverse structures and machines. Quantitative modeling has contributed to our understanding of cytoskeletal functions including mitosis (*Mogilner and Craig, 2010*; *Civelekoglu-Scholey and Cimini, 2014*), cytokinesis (*Akamatsu et al., 2014*; *Stachowiak et al., 2014*), and cell motility (*Allard and Mogilner, 2013*; *Barnhart et al., 2017*).

Chromosome segregation in eukaryotes is performed by the mitotic spindle, a self-organized microtubule (MT)-based machine (*Bray, 2000*; *McIntosh et al., 2012*). Dynamic spindle MTs are typically organized with their plus-ends toward the center of the spindle, forming a bipolar array as the spindle poles move apart (*Figure 1*; *Taylor, 1959*; *McIntosh et al., 2012*). Motor proteins and crosslinkers that bundle and slide MTs create, extend, and stabilize MT bundles (*Figure 1A,B*; *Mann and Wadsworth, 2019*; *Pidoux et al., 1996*; *Chen et al., 2012*; *Hepperla et al., 2014*; *Hueschen et al., 2019*; *Yamashita et al., 2005*; *Janson et al., 2007*; *Braun et al., 2011*; *Lansky et al., 2015*). As the spindle assembles, MTs attach to duplicated chromosomes at kinetochores and align them at the spindle midzone (*Figure 1A–C*; *Musacchio and Desai, 2017*; *Hinshaw and Harrison, 2018*; *Hamilton et al., 2019*). Biorientation occurs when sister kinetochores are attached to sister poles,

**eLife digest** Before a cell divides, it must make a copy of its genetic material and then promptly split in two. This process, called mitosis, is coordinated by many different molecular machines. The DNA is copied, then the duplicated chromosomes line up at the middle of the cell. Next, an apparatus called the mitotic spindle latches onto the chromosomes before pulling them apart. The mitotic spindle is a bundle of long, thin filaments called microtubules. It attaches to chromosomes at the kinetochore, the point where two copied chromosomes are cinched together in their middle.

Proper cell division is vital for the healthy growth of all organisms, big and small, and yet some parts of the process remain poorly understood despite extensive study. Specifically, there is more to learn about how the mitotic spindle self-assembles, and how microtubules and kinetochores work together to correctly orient and segregate chromosomes into two sister cells. These nanoscale processes are happening a hundred times a minute, so computer simulations are a good way to test what we know.

Edelmaier et al. developed a computer model to simulate cell division in fission yeast, a species of yeast often used to study fundamental processes in the cell. The model simulates how the mitotic spindle assembles, how its microtubules attach to the kinetochore and the force required to pull two sister chromosomes apart. Building the simulation involved modelling interactions between the mitotic spindle and kinetochore, their movement and forces applied. To test its accuracy, model simulations were compared to recordings of the mitotic spindle – including its length, structure and position – imaged from dividing yeast cells.

Running the simulation, Edelmaier et al. found that several key effects are essential for the proper movement of chromosomes in mitosis. This includes holding chromosomes in the correct orientation as the mitotic spindle assembles and controlling the relative position of microtubules as they attach to the kinetochore. Misaligned attachments must also be readily deconstructed and corrected to prevent any errors. The simulations also showed that kinetochores must begin to exert more force (to separate the chromosomes) once the mitotic spindle is attached correctly.

Altogether, these findings improve the current understanding of how the mitotic spindle and its counterparts control cell division. Errors in chromosome segregation are associated with birth defects and cancer in humans, and this new simulation could potentially now be used to help make predictions about how to correct mistakes in the process.

but is often preceded by erroneous attachment (*Figure 1D*; *Cimini et al., 2001*; *Salmon et al., 2005*; *Rumpf et al., 2010*; *Gregan et al., 2011*; *Lampson and Grishchuk, 2017*). Kinetochores therefore perform multiple functions: they link chromosomes to MTs, maintain attachment to MT ends under force and as MTs grow and shrink, sense MT attachment and tension between sisters, and regulate correction of attachment errors and the spindle-assembly checkpoint (*Sacristan and Kops, 2015*; *Musacchio and Desai, 2017*).

It is not fully understood how kinetochores, microtubules, and associated proteins robustly assemble a bipolar spindle and align chromosomes. In particular, it is unclear which kinetochore functions are most important for error correction and proper chromosome segregation (*Lampson and Grishchuk, 2017*; *Sacristan and Kops, 2015*). Error correction is affected by kineto-chore geometry (*Gregan et al., 2007*; *Paul et al., 2009*; *Rumpf et al., 2010*; *Magidson et al., 2015*; *Zaytsev and Grishchuk, 2015*) and attachment/tension sensing (*Sacristan and Kops, 2015*; *Musacchio, 2015*; *Musacchio and Desai, 2017*; *Salmon and Bloom, 2017*), although the relative contribution of different effects is not established (*Nannas and Murray, 2014*; *Tauchman et al., 2015*; *Kuhn and Dumont, 2017*; *Yoo et al., 2018*). Destabilization of incorrect attachments by Aurora B kinase appears to be particularly important for high-fidelity chromosome segregation (*Cheeseman et al., 2002*; *Cimini et al., 2006*; *Liu et al., 2009*; *Liu et al., 2010a*). Therefore, further insight into the minimal mechanisms required for spindle assembly and chromosome biorientation could be gained from a computational model.

Once the spindle assembles and attaches to chromosomes, it achieves a consistent length (*Dumont and Mitchison, 2009*; *Goshima and Scholey, 2010*; *Nannas et al., 2014*; *Rizk et al., 2014*; *Lacroix et al., 2018*). The force-balance model proposes that outward-directed forces from

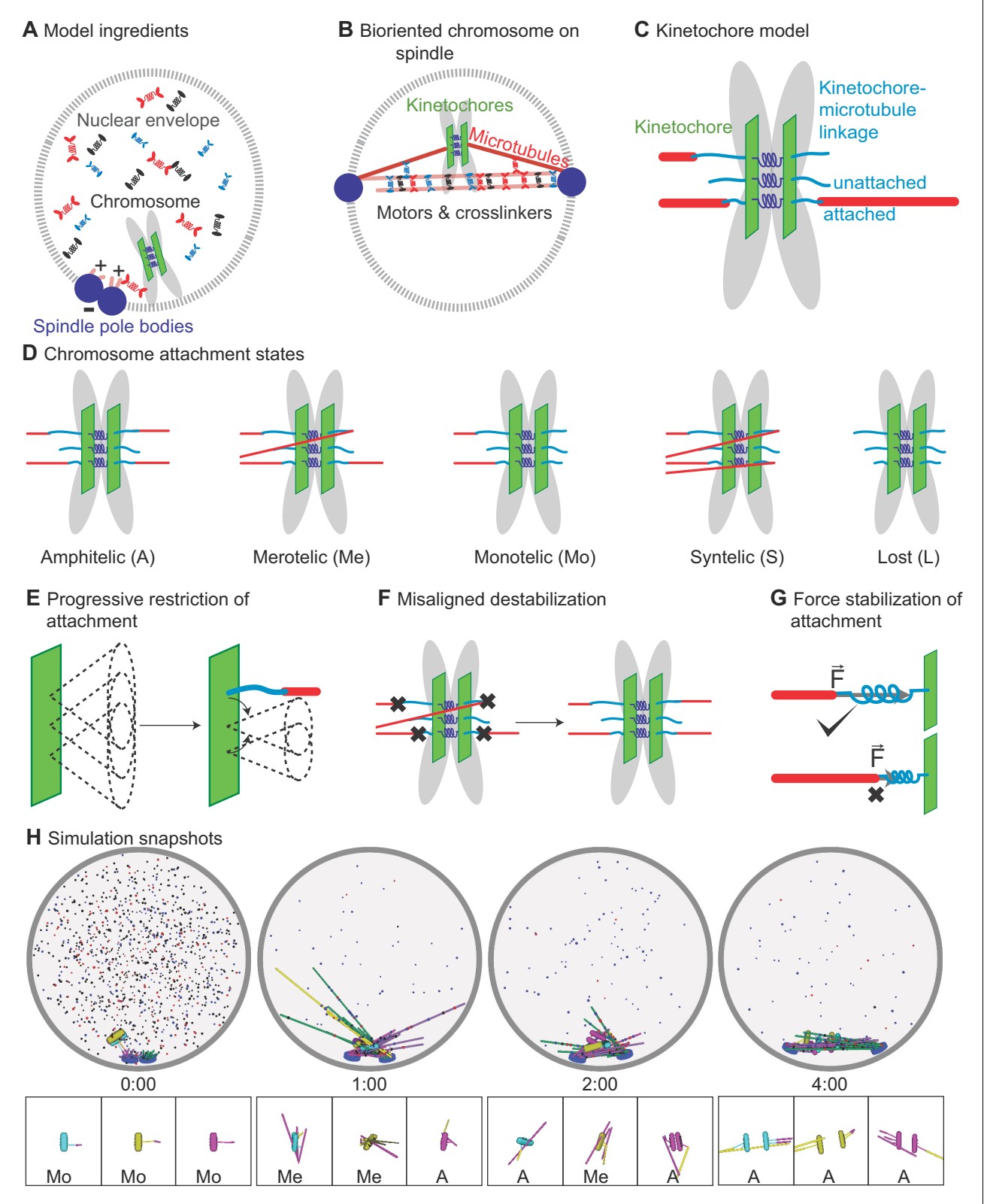

**Figure 1.** Schematic of computational model and simulation of the reference model. (**A**) Schematic of initial condition, showing adjacent spindle-pole bodies (blue) embedded in the nuclear envelope (gray dashed), proximal chromosomes (gray with green plate and blue springs), short microtubules (pink), and motor proteins and crosslinkers (red, blue, and black). (**B**) Schematic of bipolar spindle and a bioriented chromosome. (**C**) Schematic of chromosome and kinetochore model showing sister chromatids (gray), one kinetochore on each chromatid (green plates), the pericentric chromatin

*Figure 1 continued on next page*

**Figure 1 continued**

spring (blue springs), and kinetochore-MT attachment factor (blue line). (D) Schematic of chromosome attachment states, showing amphitelic, merotelic, monotelic, syntelic, and lost chromosomes. (E) Schematic of progressive restriction, showing that the angular range of kinetochore-MT attachment is restricted after attachment. (F) Schematic of misaligned destabilization of attachment, showing that misaligned attachments are destabilized. (G) Schematic of force stabilization of attachment, showing that end-on attachment to depolymerizing MTs has increased lifetime. (H) Image sequence of spindle assembly and chromosome biorientation rendered from a three-dimensional simulation. Initially, spindle-pole bodies (SPBs) are adjacent (blue disks), MTs are short spherocylinders (green and purple when unattached to kinetochores, yellow and magenta when attached), and chromosomes (cyan, yellow, magenta) are near SPBs. Motors and crosslinkers are dispersed spots (red, blue, and black) within the nucleus (gray boundary). Time shown in minutes:seconds. Lower: a zoomed view of each chromosome with attachment state labeled.

The online version of this article includes the following source data for figure 1:

**Source data 1.** Configuration files for the simulations used for snapshots in *Figure 1H*.

---

plus-end directed sliding motors separate spindle poles, while inward-directed forces from minus-end directed sliding motors and chromosomes pull the poles together (*Saunders and Hoyt, 1992*). This model helps explain perturbations that alter spindle length (*Syrovatkina et al., 2013*; *Hepperla et al., 2014*; *Chacón et al., 2014*; *Nannas et al., 2014*). However, a change in spindle length may occur from a direct change in force production or from indirect effects such as alteration in MT dynamics or alignment (*Hepperla et al., 2014*; *Gergely et al., 2016*). In addition, the steady-state force-balance model requires extension to address spindle length fluctuations, in which the bipolar spindle assembles, but then undergoes large, dynamic length changes (*Bratman and Chang, 2007*; *Griffiths et al., 2008*; *Choi et al., 2009*; *Hsu and Toda, 2011*; *Masuda et al., 2013*; *Wälde and King, 2014*; *Syrovatkina et al., 2013*; *Gergely et al., 2016*). Computational modeling can be a valuable tool to dissect force generation and spindle length changes.

To better understand the key mechanistic requirements for chromosome biorientation and how kinetochore number and attachment affect spindle length stability, we developed a computational model of fission-yeast mitosis. *Schizosaccharomyces pombe* cells are amenable to genetic manipulation and quantitative experiments (*Ward et al., 2015*; *Mary et al., 2015*; *Klemm et al., 2018*; *Blackwell et al., 2017b*; *Blackwell et al., 2017a*) and the spindles are small enough that full 3D simulations are computationally tractable (*Glunčić et al., 2015*; *Ward et al., 2015*; *Blackwell et al., 2017a*; *Lamson et al., 2019*). We were motivated by previous work modeling spindle function and chromosome segregation (*Mogilner and Craig, 2010*; *Civelekoglu-Scholey and Cimini, 2014*). Because we study de novo spindle assembly and chromosome alignment, we could not use previous models that started with an already-bipolar structure and/or chromosomes attached to the spindle. Therefore, we extended a previous model of spindle assembly in the absence of chromosomes and kinetochore-microtubule attachments (*Blackwell et al., 2017a*; *Rincon et al., 2017*; *Lamson et al., 2019*) to include chromosomes and kinetochores.

Our model successfully accomplishes spindle assembly and chromosome biorientation. The results give insight into key requirements for error correction and long-lived biorientation, emphasizing the importance of progressive restriction of attachment, destabilization of misaligned attachments, and force-dependent attachment lifetime. The turnover of kinetochore-MT attachments affects spindle mechanics, because models with larger attachment lifetime exhibit larger fluctuations in spindle length. The spindle components which contribute most to force generation change over time: initial spindle -pole separation is due to the outward force from kinesin-5 motors overcoming the passive crosslinker braking force, while interkinetochore stretch is the main inward force after biorientation. Finally, properly constructed metaphase spindles are able to robustly segregate chromosomes in the model.

## Materials and methods

Computational modeling has been used previously to study the mitotic spindle (*Mogilner and Craig, 2010*; *Vladimirou et al., 2011*; *Civelekoglu-Scholey and Cimini, 2014*). Recent work on spindle and MT organization includes studies of spindle elongation and force balance (*Brust-Mascher et al., 2015*; *Ward et al., 2015*), the formation and maintenance of antiparallel MT overlaps (*Johann et al., 2015*; *Johann et al., 2016*), MT bundling and sliding (*Hepperla et al., 2014*), spindle movements and positioning (*Ma et al., 2014*; *Garzon-Coral et al., 2016*), spindle length and shape

(*Brugués and Needleman, 2014*; *Chacón et al., 2014*; *Hepperla et al., 2014*; *Gergely et al., 2016*; *Novak et al., 2018*), MT organization (*Redemann et al., 2017*), and spindle assembly from a bipolar initial condition (*Magidson et al., 2015*; *Winters et al., 2019*). Models of kinetochore-MT attachment and biorientation have examined capture of lost kinetochores (*Kalinina et al., 2013*; *Blackwell et al., 2017b*), chromosome reorientation after MT attachment (*Paul et al., 2009*), attachment error correction (*Zaytsev and Grishchuk, 2015*; *Tubman et al., 2017*; *Yoo et al., 2018*; *Trivedi et al., 2019*), and chromosome movement on the spindle (*Armond et al., 2015*; *Banigan et al., 2015*; *Gergely et al., 2016*; *Vukušić et al., 2017*; *Klemm et al., 2018*). Most spindle models have started with a bipolar structure or separated spindle poles, and most previous chromosome models have begun with chromosomes attached to the spindle or near a pre-formed spindle. Because we seek to model simultaneous spindle assembly and chromosome alignment with few imposed constraints, we developed a new model, building on previous work on spindle assembly in the absence of chromosomes and kinetochore-microtubule attachments (*Blackwell et al., 2017a*; *Rincon et al., 2017*; *Lamson et al., 2019*).

In developing the model, we used three-dimensional geometry and an initial condition with side-by-side centrosomes (spindle-pole bodies, or SPBs) that mimics the biological configuration at the onset of mitosis. Because stochastic binding kinetics and turnover of motor proteins, crosslinkers, and kinetochore-MT attachments are important in spindle assembly and chromosome alignment, we developed methods to rapidly compute the statistical mechanics of protein binding and unbinding (*Gao et al., 2015b*; *Gao et al., 2015a*; *Blackwell et al., 2016*; *Blackwell et al., 2017a*). The binding and unbinding of motors, crosslinkers, and kinetochore-MT attachments is microscopically reversible and force-dependent. Motor proteins move with force-dependent velocity, which can be important for force generation by crosslinking motors (*Blackwell et al., 2016*; *Blackwell et al., 2017a*). We represent steric interactions between molecules (such as microtubules) with a hard-core short-range repulsive interaction, rather than soft repulsion. The simulations are based on Brownian dynamics, and state changes (such as motor binding/unbinding and an MT switching from growing to shrinking) are modeled using kinetic Monte Carlo sampling (*Blackwell et al., 2017a*; *Blackwell et al., 2017b*; *Rincon et al., 2017*; *Lamson et al., 2019*; Appendix 1, *Table 1*; *Table 2*; *Table 3*). We seek quantitative agreement between results from the simulation model and experiments, and so fix poorly constrained model parameters by direct comparison to data (*Blackwell et al., 2017a*; *Rincon et al., 2017*).

## Geometry, microtubules, motors, and crosslinkers

The simulation takes place within a sphere that represents the fission-yeast nucleus. Two SPBs are embedded in the nuclear envelope but free to move on the surface of the sphere (although we also consider effects of allowing SPBs to move radially due to a soft nuclear envelope in one variant of the model, as discussed below). Each SPB nucleates 14 MTs, with their minus-ends tethered to the SPBs by a spring and which undergo dynamic instability at their plus-ends. Steric interactions are mediated by short-range hard repulsion between MTs, SPBs, and the nuclear envelope (*Figure 1A,B*, Appendix 1).

Three classes of motors and crosslinkers assemble the spindle (*Figure 1A,B*). Kinesin-5 motors (representing Cut7) move bidirectionally on MTs (*Edamatsu, 2014*; *Edamatsu, 2016*; *Britto et al., 2016*; *Singh et al., 2018*), with plus-end directed movement on antiparallel MTs exerting force to slide apart the SPBs. Kinesin-14 motors (representing Pkl1 and Klp2) crosslink MTs and one head walks toward the MT minus-ends, aligning MTs and exerting force that shortens the spindle (*Pidoux et al., 1996*; *Troxell et al., 2001*; *Chen et al., 2012*; *Olmsted et al., 2014*; *Hepperla et al., 2014*; *Yukawa et al., 2015*; *Yukawa et al., 2018*). Crosslinkers (representing Ase1) preferentially bind antiparallel MTs (*Yamashita et al., 2005*; *Loïodice et al., 2005*; *Janson et al., 2007*; *Kapitein et al., 2008*; *Courtheoux et al., 2009*; *Fu et al., 2009*) and stabilize MT overlaps when crosslinking near the end of an MT, an effect which mimics the recruitment of stabilizing proteins such as CLASP (*Bratman and Chang, 2007*) to MT ends.

## Chromosomes and kinetochores

We represent the multiple outer kinetochore proteins involved in MT binding (*Sacristan and Kops, 2015*; *Musacchio and Desai, 2017*) by a single attachment factor that can be bound or unbound to

**Table 1.** Simulation, SPB, and MT parameters.

| Simulation parameter | Symbol | Value | Notes |
|---|---|---|---|
| Time step | $\delta t$ | $8.9 \times 10^{-6}$ s | Blackwell et al., 2017a |
| Nuclear envelope radius | $R$ | 1.375 µm | Kalinina et al., 2013 |
| **Spindle pole bodies** | | | |
| Diameter | $\sigma_{\text{SPB}}$ | 0.1625 µm | Ding et al., 1993 |
| Bridge size | | 75 nm | Ding et al., 1993 |
| Tether length | $R_0$ | 50 nm | Flory et al., 2002; Muller et al., 2005 |
| Tether spring constant | $K_0$ | 0.6625 pN nm$^{-1}$ | Blackwell et al., 2017a |
| Translational diffusion coefficient | $D_t$ | $4.5 \times 10^{-4}$ µm$^2$ s$^{-1}$ | Blackwell et al., 2017a |
| Rotational diffusion coefficient | $D_{\theta,\text{spb}}$ | 0.0170 s$^{-1}$ | Blackwell et al., 2017a |
| Linkage time | $\tau_{link}$ | 5 s | Blackwell et al., 2017a |
| **Microtubules** | | | |
| Diameter | $\sigma_{MT}$ | 25 nm | Blackwell et al., 2017a |
| Angular diffusion coefficient | $D_\theta$ | Depends on MT length | Blackwell et al., 2017a; Kalinina et al., 2013 |
| Force-induced catastrophe constant | $\alpha_c$ | 0.5 pN$^{-1}$ | Blackwell et al., 2017a; Janson et al., 2003; Dogterom and Yurke, 1997 |
| Growth speed | $v_{p,0}$ | 4.1 µm min$^{-1}$ | Blackwell et al., 2017a; Blackwell et al., 2017b |
| Shrinking speed | $v_{s,0}$ | 6.7 µm min$^{-1}$ | Blackwell et al., 2017a; Blackwell et al., 2017b |
| Catastrophe frequency | $f_{c,0}$ | 3.994 min$^{-1}$ | Blackwell et al., 2017a; Blackwell et al., 2017b |
| Rescue frequency | $f_{r,0}$ | 0.157 min$^{-1}$ | Blackwell et al., 2017a; Blackwell et al., 2017b |
| Growth speed stabilization | $s_{vg}$ | 1.54 | Optimized |
| Shrinking speed stabilization | $s_{vs}$ | 0.094 | Optimized |
| Catastrophe frequency stabilization | $s_{fc}$ | 0.098 | Optimized |
| Rescue frequency stabilization | $s_{fr}$ | 18 | Optimized |
| Stabilization length | $s_\ell$ | 16 nm | Optimized |
| Minimum MT length | $L_{\min}$ | 75 nm | Optimized |

an MT. Because fission-yeast kinetochores can bind up to 3 MTs (*Ding et al., 1993*), each kineto-chore has three attachment factors in the model separated by 40 nm along the kinetochore plate (*Figure 1C*, *Appendix 1—figure 1*). Attachments are constrained so that no more than one attach-ment factor can bind to the same MT plus-end. The attachment factor is a 54-nm-long spring that exerts force on the MT and kinetochore when stretched or compressed (*Tables 4* and *5*). Attachment factors can make both lateral and end-on attachments to MTs, with different binding kinetics that favor end-on attachment. Importantly, the model includes tip tracking: a tip-bound attachment

**Table 2.** Soft nuclear envelope model parameters.

| Parameter | Symbol | Value | Notes |
|---|---|---|---|
| Translational mobility | $\mu^{tb}_{\text{SPB}}$ | $\begin{pmatrix} 0.05 & 0 & 0 \\ 0 & 0.11 & 0 \\ 0 & 0 & 0.11 \end{pmatrix} \mu \text{ms}^{-1}\,\text{pN}^{-1}$ | Calculated |
| Rotational mobility | $\mu^{rb}_{\text{SPB}}$ | $\begin{pmatrix} 16.6 & 0 & 0 \\ 0 & 0.166 & 0 \\ 0 & 0 & 0.166 \end{pmatrix} \mu \text{m}^{-1}\text{s}^{-1}\text{pN}^{-1}$ | Calculated |
| Membrane tube radius | $f_{\text{tube}}$ | 87.7 nm | Derényi et al., 2002; Lim et al., 2007; Lamson et al., 2019 |
| MT asymptotic wall force | $f_{\text{MT,w}}$ | 2.5 pN | Derényi et al., 2002; Lim et al., 2007; Lamson et al., 2019 |
| SPB asymptotic wall force | $f_{\text{SPB,w}}$ | 17 pN | Derényi et al., 2002; Lim et al., 2007; Lamson et al., 2019 |
| Tether spring constant | $K_0$ | 6.625 pN nm$^{-1}$ | Optimized |

**Table 3.** Motor and crosslinker parameters.

| Simulation parameter | Symbol | Value | Notes |
|---|---|---|---|
| **Kinesin-5** | | | |
| Number | $N_{K5}$ | 174 | Optimized (*Carpy et al., 2014*) |
| Association constant per site | $K_a$ | 90.9 µM$^{-1}$ site$^{-1}$ | *Cochran et al., 2004* |
| One-dimensional effective concentration | $c_2$ | 0.4 nm$^{-1}$ | *Blackwell et al., 2017a* |
| Spring constant | $K$ | 0.3 pN nm$^{-1}$ | *Kawaguchi and Ishiwata, 2001* |
| Singly-bound velocity | $v_0$ | $-100$ nm s$^{-1}$ | *Roostalu et al., 2011* |
| Polar aligned velocity | $v_{0,P}$ | $-50$ nm s$^{-1}$ | *Gerson-Gurwitz et al., 2011* |
| Anti-polar aligned velocity | $v_{0,AP}$ | 8 nm s$^{-1}$ | *Gerson-Gurwitz et al., 2011* |
| Singly bound off-rate | $k_1$ | 0.11 s$^{-1}$ | *Roostalu et al., 2011* |
| Doubly bound off-rate (single head) | $k_2$ | 0.055 s$^{-1}$ | *Blackwell et al., 2017a* |
| Tether length | $R_0$ | 53 nm | *Kashlna et al., 1996* |
| Stall force | $F_s$ | 5 pN | *Valentine et al., 2006* |
| Characteristic distance | $x_c$ | 1.5 nm | Optimized (*Arpağ et al., 2014* |
| Diffusion constant (solution) | $D_{\text{free}}$ | 4.5 µm$^2$ s$^{-1}$ | *Bancaud et al., 2009* |
| **Kinesin-14** | | | |
| Number | $N_{K14}$ | 230 | Optimized (*Carpy et al., 2014*) |
| Association constant (motor head) | $K_{a,m}$ | 22.727 µM$^{-1}$ site$^{-1}$ | *Chen et al., 2012* |
| Association constant (passive head) | $K_{a,d}$ | 22.727 µM$^{-1}$ site$^{-1}$ | *Blackwell et al., 2017a* |
| 1D effective concentration (motor head) | $c_{2m}$ | 0.1 nm$^{-1}$ | *Blackwell et al., 2017a* |
| 1D effective concentration (passive head) | $c_{2d}$ | 0.1 nm$^{-1}$ | *Blackwell et al., 2017a* |
| Spring constant | $K$ | 0.3 pN nm$^{-1}$ | *Kawaguchi and Ishiwata, 2001* |
| Singly bound velocity (motor head) | $v_{0m}$ | $-50$ nm s$^{-1}$ | *Blackwell et al., 2017a* |
| Diffusion constant (bound, diffusing head) | $D_d$ | 0.1 µm$^2$ s$^{-1}$ | *Blackwell et al., 2017a* |
| Singly bound off-rate (motor head) | $k_{1m}$ | 0.11 s$^{-1}$ | *Blackwell et al., 2017a* |
| Singly bound off-rate (passive head) | $k_{1d}$ | 0.1 s$^{-1}$ | *Blackwell et al., 2017a* |
| Doubly bound off-rate (motor head) | $k_{2m}$ | 0.055 s$^{-1}$ | *Blackwell et al., 2017a* |
| Doubly bound off-rate (passive head) | $k_{2d}$ | 0.05 s$^{-1}$ | *Blackwell et al., 2017a* |
| Tether length | $R_0$ | 53 nm | *Blackwell et al., 2017a* |
| Stall force | $F_s$ | 5.0 pN | *Blackwell et al., 2017a* |
| Characteristic distance | $x_c$ | 4.8 nm | Optimized (*Arpağ et al., 2014*) |
| Adjusted characteristic distance | $x_c'$ | 1.5 nm | *Figure 2—figure supplement 1C* |
| **Crosslinker** | | | |
| Number | $N_{XL}$ | 657 | Optimized (*Carpy et al., 2014*) |
| Association constant | $K_a$ | 90.9 µM$^{-1}$ site$^{-1}$ | *Cochran et al., 2004* |
| One-dimensional effective concentration | $c_2$ | 0.4 nm$^{-1}$ | *Lansky et al., 2015* |
| Spring constant | $K$ | 0.207 pN nm$^{-1}$ | *Lansky et al., 2015* |
| Diffusion constant (solution) | $D_{\text{free}}$ | 4.5 µm$^2$ s$^{-1}$ | *Bancaud et al., 2009* |
| Singly bound diffusion constant | $D_{\text{sb}}$ | 0.1 µm$^2$ s$^{-1}$ | *Lansky et al., 2015* |
| Doubly bound diffusion constant | $D_{\text{db}}$ | $6.7 \times 10^{-3}$ µm$^2$ s$^{-1}$ | *Lansky et al., 2015* |
| Singly bound off-rate | $k_1$ | 0.1 s$^{-1}$ | *Kapitein et al., 2008* |
| Doubly bound off-rate | $k_2$ | 0.05 s$^{-1}$ | *Lansky et al., 2015* |
| Parallel-to-antiparallel bindng ratio | $P_{\text{aff}}$ | 0.33 | *Kapitein et al., 2008*; *Rincon et al., 2017*; *Lamson et al., 2019* |
| Characteristic distance | $x_c$ | 2.1 nm | Optimized (*Arpağ et al., 2014*) |
| Tether length | $R_0$ | 53 nm | *Lansky et al., 2015*; *Lamson et al., 2019* |

**Table 4.** Chromosome and kinetochore parameters.

| Simulation parameter | Symbol | Value | Notes |
|---|---|---|---|
| **Kinetochore kinematics** | | | |
| Diameter | $\sigma_{KC}$ | 200 nm | *Blackwell et al., 2017a*; *Kalinina et al., 2013* |
| Length | $L_{KC,0}$ | 150 nm | *Ding et al., 1993* |
| Width | $L_{KC,1}$ | 50 nm | *Ding et al., 1993* |
| Thickness | $d_{KC}$ | 0 nm | Chosen |
| Diffusion coefficient | $D_{KC}$ | $5.9 \times 10^{-4} \mu m^2\ s^{-1}$ | *Gergely et al., 2016*; *Blackwell et al., 2017a*; *Kalinina et al., 2013* |
| Translational drag | $\gamma_{KC,t}$ | 3.51 pN $\mu m^{-1}$ s | Computed |
| Rotational drag | $\gamma_{KC,r}$ | 0.165 pN $\mu m$ s | Computed |
| Catastrophe enhancement | $s_{KC-cen,fc}$ | 0.5 pN$^{-1}$ | Matches NE factor |
| MT tip length | $l_{cen,tip}$ | 25 nm | Chosen |
| **Interkinetochore spring** | | | |
| Tether length | $R_{C,0}$ | 100 nm | *Stephens et al., 2013*; *Gergely et al., 2016*; *Gay et al., 2012* |
| Linear spring constant | $\kappa_C$ | 39 pN $\mu m^{-1}$ | Optimized |
| Rotational spring constant | $\kappa_{C,u}$ | 1850 pN nm rad$^{-1}$ | Optimized |
| Alignment spring constant | $\kappa_{C,v}$ | 1850 pN nm rad$^{-1}$ | Optimized |
| **Pericentric chromatin** | | | |
| Pericentric chromatin length | $r_{centromere}$ | 200 nm | Chosen |
| Pericentric chromatin diameter | $d_{centromere}$ | 75 nm | Chosen |
| Kinetochore-centromere offset | $r_{KC-cen}$ | 37.5 nm | Chosen |
| Chromatin-MT repulsion amplitude | $A_{CMT}$ | 1 pN nm | Optimized |

factor tracks MT ends by maintaining end-on attachment during MT growth and shrinking. The attachment factor also includes a plus-end-directed kinetochore motor, representing the measured contribution of kinetochore-localized dimeric Cut7 to chromosome alignment (*Akera et al., 2015*). End-on attachment alters MT dynamic instability and is force-dependent, as measured previously (*Akiyoshi et al., 2010*; *Miller et al., 2016*).

Physically each kinetochore is a rectangular plate of length 150 nm, width 50 nm, and zero thickness (*Figure 1C*; *Ding et al., 1993*) with a steric repulsion with MTs. Sister kinetochores are linked via springs that resist stretching and rotation, to maintain the distance and alignment of the kinetochores (*Figure 1C*, *Appendix 1—figure 1*; *Mary et al., 2015*; *Smith et al., 2016*). The pericentric DNA is modeled as a spherocylinder of length 200 nm and diameter 75 nm, which has a soft repulsion with MTs that allows MT-chromatin overlap with an energy penalty (Appendix 1).

With these ingredients, the model can achieve both correct and erroneous kinetochore-MT attachment states (*Figure 1D*). To achieve error correction and persistent biorientation, we found three key model ingredients were required: progressive restriction of attachment (*Figure 1E*), destabilization of misaligned attachment (*Figure 1F*), and stabilization of attachment by force (*Figure 1G*, Appendix 1). With these mechanisms, the model exhibits both spindle assembly and chromosome biorientation (*Figure 1H*, *Video 1*).

## Comparison to experimental results

To constrain model parameters, we developed multiple tests of simulation performance based on live-cell imaging, electron microscopy, and biorientation. First, we quantified the dynamics of spindle length and kinetochore position by confocal fluorescence light microscopy (*Figure 2*; *Gergely et al., 2016*; *Blackwell et al., 2017a*). Cells with low-level labeling of MTs with *mCherry-atb2* (*Yamagishi et al., 2012*; *Blackwell et al., 2017a*) and the *cen2-GFP* marker on the centromeric DNA of chromosome 2 (*Yamamoto and Hiraoka, 2003*) allowed imaging of spindle length and centromere position (Appendix 1). The Cen2 marker is displaced only 125 nm on average from the kinetochore (*Gay et al., 2012*), allowing quantification of the position of a single pair of sister

**Table 5.** Attachment factor parameters.

| Parameter | Symbol | Value | Notes |
|---|---|---|---|
| Number | $N_{AF}$ | 3 | *Ding et al., 1993* |
| Attachment-site separation on kinetochore | $r_{AF,ex}$ | 40 nm | *Ding et al., 1993* |
| Linear spring constant | $\kappa_{AF,m}$ | 0.088 pN nm$^{-1}$ | Optimized |
| Angular spring constant, 0 to 1 | $\kappa_{AF,r,0}$ | 4.1 pN nm | Optimized |
| Angular spring constant, 1 to 2 | $\kappa_{AF,r,1}$ | 41 pN nm | Optimized |
| Angular spring constant, 2 to 3 | $\kappa_{AF,r,2}$ | 410 pN nm | Optimized |
| Angular spring constant, 3 to 3 | $\kappa_{AF,r,3}$ | 410 pN nm | Optimized |
| Tether length | $r_{AF,0}$ | 54 nm | *Ciferri et al., 2007* |
| kMC steps | $N_{kmc}$ | 10 | Chosen |
| MT tip length | $l_{AF,tip}$ | 25 nm | Chosen |
| MT tip crowding | $b_{AF,tip}$ | True | *Ding et al., 1993* |
| Tip concentration | $c_{AF,tip}$ | 40 nm$^{-1}$ | Optimized |
| Side concentration | $c_{AF,side}$ | 0.4 nm$^{-1}$ | Optimized |
| Tip rate assembling | $k_{AF,tip,a}$ | 0.0001 s$^{-1}$ | Optimized |
| Tip rate disassembling | $k_{AF,tip,d}$ | 0.03 s$^{-1}$ | Optimized |
| Side rate | $k_{AF,side}$ | 0.03 s$^{-1}$ | Optimized |
| Tip characteristic distance assembling | $x_{c,t,a}$ | 1 nm | Optimized |
| Tip characteristic distance disassembling | $x_{c,t,d}$ | −3.9 nm | Optimized |
| Side characteristic distance | $x_{c,s}$ | −0.37 nm | Optimized |
| Angular characteristic factor | $\chi_c$ | 0.013 | Optimized |
| Speed | $v_{AF}$ | 50 nm s$^{-1}$ | Optimized |
| Stall force | $f_{AF,stall}$ | 5 pN | Kinesin-5 (*Blackwell et al., 2017a*; *Akera et al., 2015*) |
| Tip diffusion | $D_{tip}$ | 0.0012 μm$^2$ s$^{-1}$ | Optimized |
| Side diffusion | $D_{side}$ | 0.018 μm$^2$ s$^{-1}$ | Optimized |
| Tip tracking | $f_{AF,track}$ | 0.25 | Optimized |
| Tip-enhanced catastrophe | $s_{fc,dam1}$ | 4 | Optimized |
| Misaligned destabilization | $s_{k,ABK}$ | 70 | Optimized |
| Polymerization force factor | $F_{AF,vg}$ | 8.4 pN | *Akiyoshi et al., 2010*; *Gergely et al., 2016* |
| Depolymerization force factor | $F_{AF,vs}$ | −3.0 pN | *Akiyoshi et al., 2010*; *Gergely et al., 2016* |
| Catastrophe force factor | $F_{AF,fc}$ | −2.3 pN | *Akiyoshi et al., 2010*; *Gergely et al., 2016* |
| Rescue force factor | $F_{AF,fr}$ | 6.4 pN | *Akiyoshi et al., 2010*; *Gergely et al., 2016* |
| Maximum polymerization speed | $v_{AF,MT,max}$ | 30 μm min$^{-1}$ | *Gergely et al., 2016* |

kinetochores. We measured spindle length and kinetochore position by fitting Gaussian spots and lines to detect features, and then tracked spindle length and kinetochore position over time using previous methods (Appendix 1; *Jaqaman et al., 2008*). Second, we used previously published electron tomographic reconstructions of fission yeast spindles (*Grishchuk and McIntosh, 2006*; *McIntosh et al., 2013*) to measure spindle structure (*Blackwell et al., 2017a*). Third, we quantified how successfully the models biorient chromosomes, measured by the fraction of simulation time during which all the chromosomes are bioriented and the average number of end-on attachments.

We combined these measures of simulation performance in a fitness function which quantifies the overall success of each simulation run with a set of model parameters. We then varied poorly constrained model parameters to maximize the fitness function. The optimized parameters defined the reference model (Appendix 1).

## Reference Model

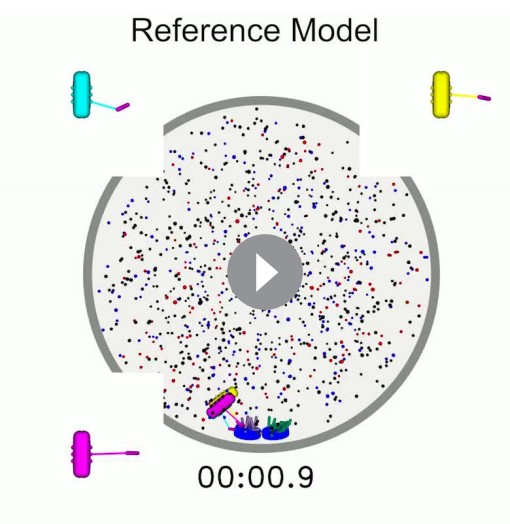

00:00.9

**Video 1.** Simulation of the reference model shows spindle assembly simultaneous with chromosome biorientation. Initially, short MTs begin to grow at the start of the simulation and interact with nearby kinetochores. A bipolar spindle forms as the chromosomes begin to biorient. Finally, a metaphase spindle is established with bioriented chromosomes that move along the spindle and breathe. The insets are zoomed views of each chromosome, showing attachment turnover and interkinetochore stretch.
https://elifesciences.org/articles/48787#video1

## Results

### A computational model can assemble a bipolar spindle and biorient chromosomes

To understand the mechanisms most important for proper chromosome alignment on the spindle, we developed a computational model of fission-yeast mitosis (*Figure 1*) that includes spindle MTs nucleated from SPBs, crosslinking motors, passive crosslinkers, pericentric chromatin, and kinetochores, all contained within a spherical nucleus (Materials and methods, *Figure 1A,B*). Kinetochore-MT binding occurs via attachment factors that represent MT-binding kinetochore proteins (*Figure 1C*), which can form both correct and erroneous MT-kinetochore attachments (*Figure 1D*). Kinetochore-MT attachments progressively restrict in angle as MTs bind (*Figure 1E*), a mechanism motivated by previous work on kinetochore geometry and chromosome rotation in error correction (*Gregan et al., 2007*; *Rumpf et al., 2010*; *Paul et al., 2009*; *Magidson et al., 2015*; *Zaytsev and Grishchuk, 2015*). In particular, work on the *S. pombe* monopolin complex has proposed that monopolin acts as a site-clamp that co-orients MTs bound to the same kinetochore (*Gregan et al., 2007*). To correct attachment errors, we included destabilization of improper attachments and tip-enhanced catastrophe (*Figure 1F*), mimicking the effectsof Aurora B kinase (*DeLuca et al., 2006*; *Cimini et al., 2006*; *Gay et al., 2012*) and recapture of lost kinetochores by MT depolymerization (*Grishchuk and McIntosh, 2006*; *Franco et al., 2007*; *Gachet et al., 2008*; *Gao et al., 2010*; *Gergely et al., 2016*). To maintain biorientation, we implemented force-dependent kinetochore-MT attachment kinetics (*Figure 1G*), based on previous work that demonstrated an increase in attachment lifetime with tension when kinetochores are attached to depolymerizing MTs (*Akiyoshi et al., 2010*; *Miller et al., 2016*). For further details of the construction of the model, see Materials and methods and Appendix 1. With these ingredients, the model is able to spontaneously assemble a bipolar spindle starting with side-by-side SPBs, form MT-kinetochore attachments, correct attachment errors, and biorient the chromosomes (*Figure 1H*, *Video 1*).

To refine and test the model, we measured spindle assembly and chromosome alignment in fission yeast (*Figure 2*, Materials and methods, Appendix 1). We quantified spindle length, SPB-kinetochore separation, and interkinetochore stretch from the onset of mitosis until chromosome segregation (*Figure 2A–D*) and used these data to adjust model parameters (Materials and methods, Appendix 1). After refinement, simulations of the reference model showed dynamics of SPB separation, kinetochore movement along the spindle, and interkinetochore stretch similar to the experimental data (*Figure 2E–H*, *Video 2*). As occurs in cells, the dynamics varied from simulation to simulation, but were similar on average (*Figure 2I*, *Appendix 1—figure 2*).

### Single model perturbations recapitulate the requirement for kinesin-5 motors and CLASP

After developing the reference model, we verified that single model perturbations recapitulate results from fission-yeast genetics. Kinesin-5 motors are essential for spindle assembly in *S. pombe*, and temperature-sensitive mutants of the kinesin-5/Cut7 fail to separate spindle-pole bodies

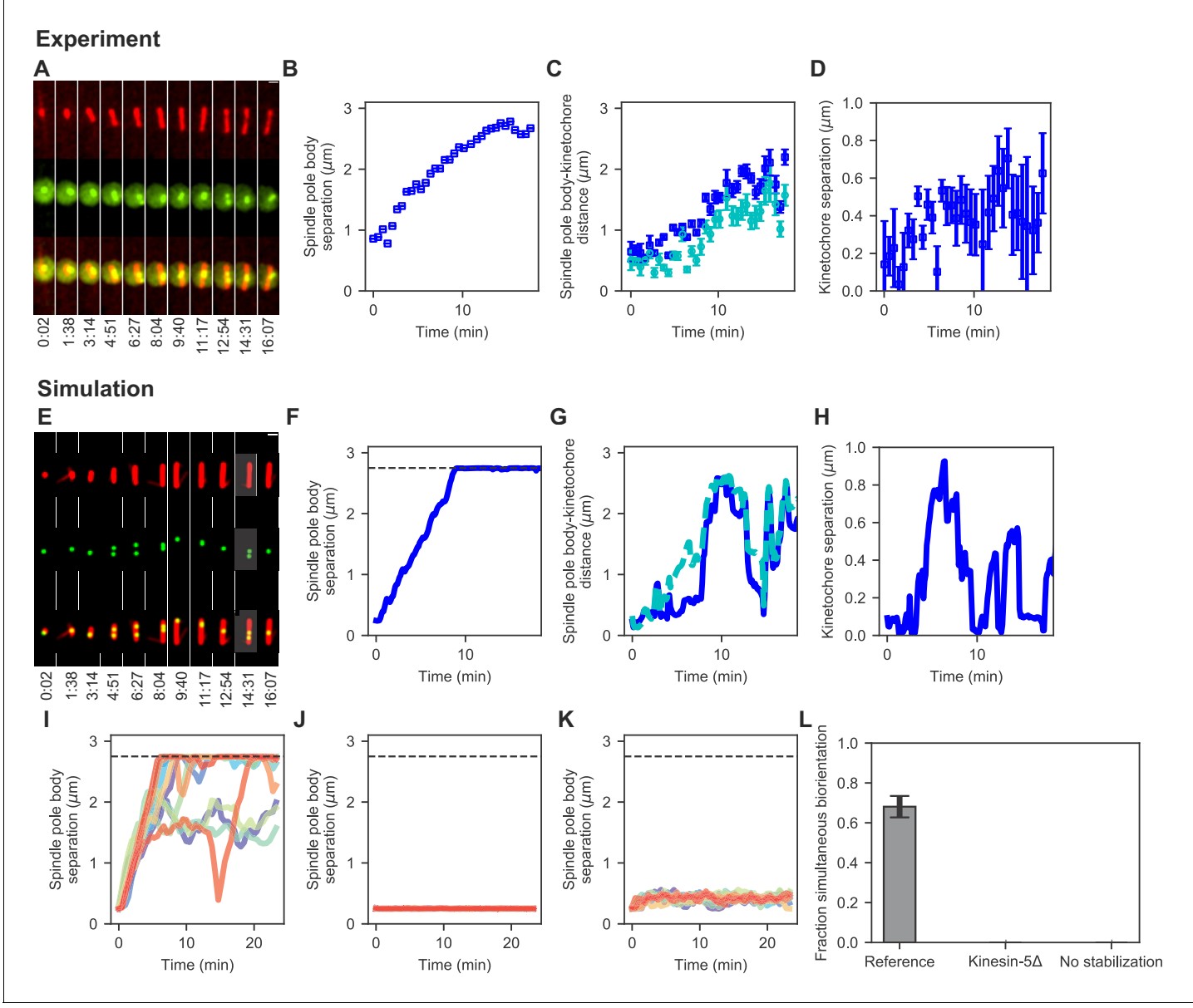

**Figure 2.** Comparison of spindle assembly and chromosome alignment in cells and simulations. (A–D) Experimental results. (A) Maximum-intensity-projected smoothed images from time-lapse confocal fluorescence microscopy of fission yeast with *mCherry-atb2* labeling MTs (red) and *cen2-GFP* labeling the centromere of chromosome 2 (green). Time shown in minutes:seconds. (B) Spindle length, (C) spindle pole body-kinetochore distance, and (D) interkinetochore distance versus time for the experiment shown in (A). (E–K) Simulation results. (E) Simulated fluorescence microscopy images with MTs (red) and a single kinetochore pair (green). (F) Spindle length, (G) spindle pole body-kinetochore distance, and (H) interkinetochore distance versus time from the simulation shown in (E), sampled at a rate comparable to the experimental data in (A–D). Note that the rigid nucleus in our model sets an upper limit on spindle length of 2.75 μm, as shown by the dashed line in F. (I) Spindle length versus time for 12 simulations of the reference model. (J) Spindle length versus time for 12 simulations in a model lacking kinesin-5. (K) Spindle length versus time for 12 simulations in a model lacking crosslink-mediated microtubule stabilization. (L) Fraction of simultaneous biorientation for the reference, kinesin-5 delete, and no-stabilization models (N = 12 simulations per data point).

The online version of this article includes the following source data and figure supplement(s) for figure 2:

**Source data 1.** Configuration and data files for the simulations used in *Figure 2*.

**Figure supplement 1.** Results of simulations with perturbations to motor and crosslinker number, motor force-dependent unbinding, and nuclear envelope rigidity.

**Figure supplement 1—source data 1.** Configuration and data files for simulations used in *Figure 2—figure supplement 1*.

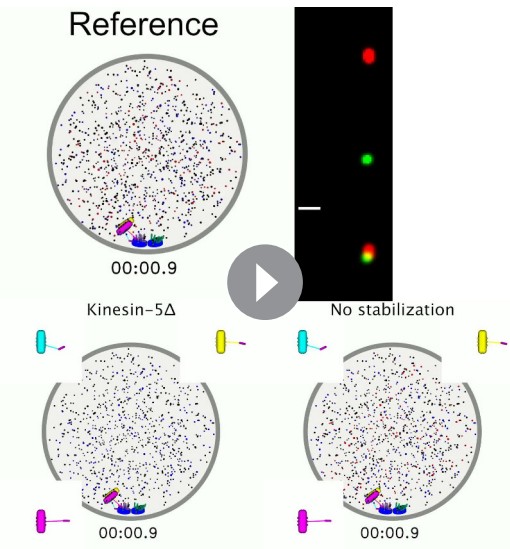

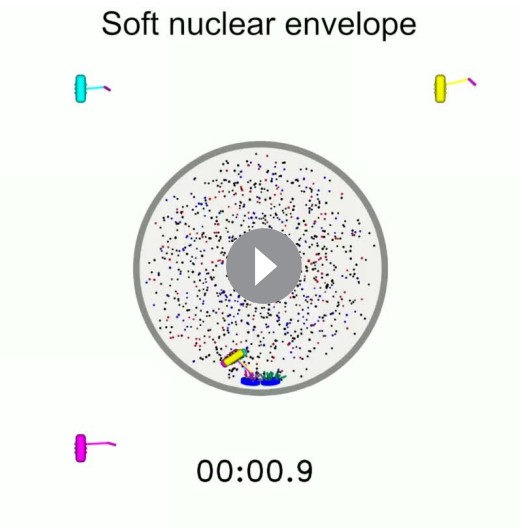

**Video 2.** Top: Simulation of reference model (left) and simulated fluorescence microscopy images (right), with red MTs and green kinetochore (scale bar 1 μm). The simulated fluorescence images are rotated so that the spindle is vertical. Lower: simulation of models mimicking genetic perturbation. Lower left: Model lacking kinesin-5 motors. The SPBs never separate and the spindle remains monopolar. Chromosomes do not biorient. Lower right: Model lacking crosslinker-mediated stabilization of MT dynamics. SPBs separate only slightly, forming a short spindle that is nearly indistinguishable from a monopolar spindle. Chromosomes do not biorient.
https://elifesciences.org/articles/48787#video2

**Video 3.** Simulation of a model with a soft nuclear envelope and an asymptotic wall force on the SPBs of 17 pN. SPBs are able to move away from their preferred radius from the center of the nucleus. The spindle reaches a bounded length, and chromosomes are able to biorient. Spindle length larger than the nuclear envelope radius is reached by the balance of force from motors, crosslinkers, chromosomes.
https://elifesciences.org/articles/48787#video3

(*Hagan and Yanagida, 1990*; *Hagan and Yanagida, 1992*; *Yukawa et al., 2018*; *Toda et al., 2018*). Consistent with this, when we remove kinesin-5 from the model, SPBs do not separate (*Figure 2J*). Similarly, the microtubule-associated protein CLASP is essential for spindle assembly in fission yeast, where it is recruited to MT antiparallel overlaps by Ase1 and stabilizes MT dynamics (*Bratman and Chang, 2007*). When the stabilization of dynamics of crosslinked MTs is turned off in the model, SPBs do not separate (*Figure 2K*). Chromosome biorientation is abolished in models where the SPBs do not separate (*Figure 2L*, *Video 2*).

We further studied combined perturbations (*Figure 2—figure supplement 1*) by varying kinesin-5 and crosslinker number in the absence of kinesin-14 (*Figure 2—figure supplement 1A*) and by varying kinesin-5 and −14 number in the absence of crosslinkers (*Figure 2—figure supplement 1B*). Kinesin-14 in our models combines the functions of fission-yeast Pkl1 and Klp2, neglecting the anchoring of MT minus-ends to SPBs by Pkl1 previously measured (*Olmsted et al., 2014*; *Syrovatkina and Tran, 2015*; *Yukawa et al., 2015*; *Yukawa et al., 2018*). Experimentally, cells lacking Klp2 or both Pkl1 and Klp2 do not show altered average spindle length (*Syrovatkina et al., 2013*; *Troxell et al., 2001*). Consistent with this, model spindles formed and bioriented chromosomes in the absence of kinesin-14, and spindle length depended on the ratio of kinesin-5 to crosslinkers.

In fission yeast, Ase1 deletion cells assemble spindles (*Yamashita et al., 2005*; *Syrovatkina et al., 2013*; *Yukawa et al., 2019*). To test if our model correctly reproduced these results, we removed the crosslinker from the model and varied the number of kinesin-5 and kinesin-14 molecules present (*Figure 2—figure supplement 1B*). Removing crosslinkers in the reference model abolished spindle assembly because spindles cannot maintain robust antiparallel MT overlaps. However, in the reference model the kinesin-14 motors are highly sensitive to force-dependent unbinding: the characteristic distance that quantifies this is 3.2 times larger for kinesin-14 motors than kinesin-5 motors. This leads to kinesin-14 motors that unbind relatively easily under force, and

they fail to maintain microtubule antiparallel overlaps necessary for bipolar spindle assembly. When we model the kinesin-14 motors with the same force sensitivity to unbinding as for the kinesin-5 motors, spindle formation and chromosome biorientation are rescued (*Figure 2—figure supplement 1C*).

Most of our simulations represent the nuclear envelope as a rigid sphere with the SPBs constrained to move on the surface of this sphere. However, constraining SPBs to a fixed radius alters force balance on the spindle and may alter spindle length. Therefore, we tested a model of a soft nuclear envelope by allowing the SPBs to move radially in a potential that mimics the energy required to deform the nuclear envelope (*Rincon et al., 2017*; *Lamson et al., 2019*) (Materials and methods, Appendix 1). The results show that a soft nuclear envelope leads to slightly longer spindles (*Figure 2—figure supplement 1D*, *Video 3*), but for a physically realistic nuclear envelope force of around 17 pN, spindle length remains near 3 µm, as measured experimentally.

## Chromosome biorientation during spindle assembly requires three basic kinetochore properties

Our simulations start in a state mimicking early mitosis with monotelic chromosomes, then spontaneously assemble a bipolar spindle and biorient chromosomes. Biorientation requires the model to correct attachment errors and maintain correct attachments. This occurs in the simulations primarily through progressive restriction of attachment angle, misaligned destabilization, and force-dependent kinetochore-MT attachment.

## Kinetochores can avoid merotelic attachments by progressive restriction of microtubule binding

To facilitate correct initial attachment of MTs to kinetochores, the model progressively restricts the angle at which binding can occur as more MTs bind (*Figure 1E*). This is motivated by previous work demonstrating that kinetochore geometry and chromosome rotation play an important role in promoting correct kinetochore-MT attachment and correcting errors (*Gregan et al., 2007*; *Rumpf et al., 2010*; *Paul et al., 2009*; *Magidson et al., 2015*; *Zaytsev and Grishchuk, 2015*). We have extended previous work to include both multiple MT binding sites per kinetochore and changes in kinetochore geometry upon binding. In our model, unattached kinetochores have a wide angular range over which attachments can form (modeled as an angular spring constant for binding, represented by the three wide cones in *Figure 1E* left). Each attachment formed narrows the angle allowed for the subsequent attachment, favoring attachment to MTs that are more perpendicular to the kinetochore plate (represented by the narrower cones in *Figure 1E* right). Attachments exert an alignment force/torque on kinetochores and MTs based on the stiffness of this angular spring.

To illustrate the importance of progressive restriction, we removed it, making the angular range identical for all three kinetochore-MT attachment events (*Figure 3A*, *Video 4*). Doing this nearly abolishes biorientation in the model: the fraction of simulation time for which all three chromosomes are bioriented is below 10%, independent the value of the angular spring constant from 1 $k_BT$ (almost any angle of attachment is allowed) to 100 $k_BT$ (attachment is highly restricted in angle). These failures occur for different reasons as the angular spring constant varies. When attachment angle is most permissive, merotelic attachments form and are not corrected sufficiently rapidly to biorient the chromosomes. When the attachment angle is highly restricted, attachments are unlikely to form at all. Overall, this result shows that in our model progressive restriction of attachment is essential for biorientation.

The progressive restriction model requires that the first binding event be relatively permissive in angle, the second more restricted, and the third highly restricted. To study this, we varied the angular spring constant of each attachment independently (*Figure 3B,C*, *Figure 3—figure supplement 1*, *Video 4*). The model achieves a high fraction of simultaneous biorientation around 70% when the first attachment is maximally permissive (spring constant is 1 $k_BT$); an increase in this spring constant restricts the angle and decreases simultaneous biorientation to below 20% (*Figure 3B*). This means that for the first attachment, promoting kinetochore binding to any MT is important: initial attachments should be established easily, even if erroneous. By contrast, biorientation is increased when the third (final) binding event is highly restricted (*Figure 3C*): chromosomes are bioriented in the model <10% of the time when the third attachment is most permissive, but the fraction of

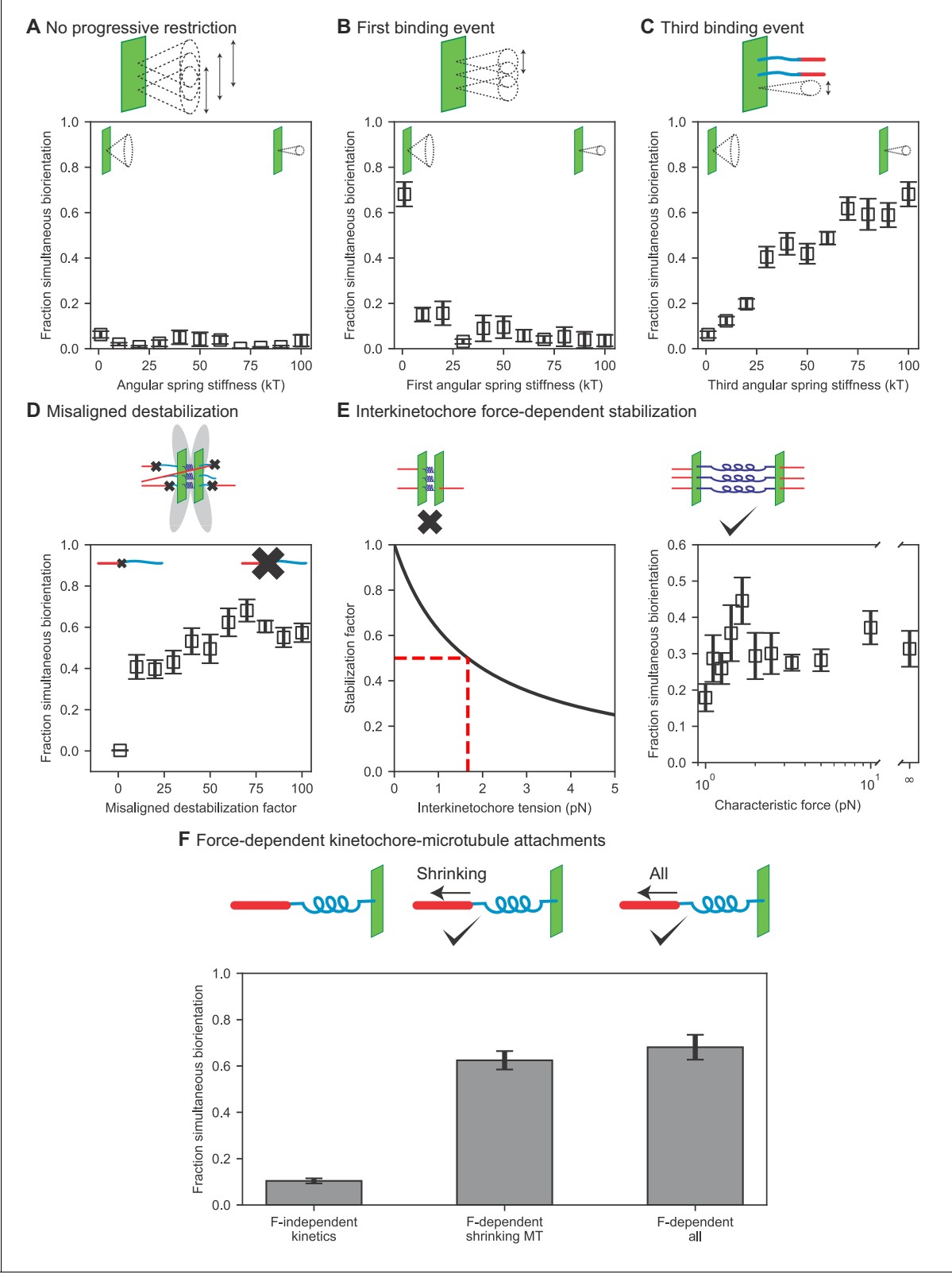

**Figure 3.** Results of perturbing kinetochore properties required for biorientation. (**A**) Fraction simultaneous biorientation versus angular spring stiffness in models lacking progressive restriction of attachment. (**B**) Fraction simultaneous biorientation versus the first angular spring stiffness in the model with progressive restriction. (**C**) Fraction simultaneous biorientation versus the third angular spring stiffness in the model with progressive restriction. (**D**) Fraction simultaneous biorientation versus the misaligned destabilization factor. (**E**) Effects of force-dependent error correction. Top, schematic of

*Figure 3 continued on next page*

*Figure 3 continued*

stabilization of kinetochore-MT attachments as a function of interkinetochore force. Left, Stabilization as a function of interkinetochore tension for a characteristic force of 1.67 pN. When the interkinetochore force is the characteristic force, attachment turnover is reduced by a factor of two, as shown by the red dashed lines. Right, fraction simultaneous biorientation versus the characteristic force. (F) Fraction simultaneous biorientation for different types of force-dependent kinetics (N = 12 simulations per data point).

The online version of this article includes the following source data and figure supplement(s) for figure 3:

**Source data 1.** Configuration and data files for simulations used in *Figure 3*.

**Figure supplement 1.** Effects of varying the middle angular stiffness for progressive restriction and the number of kinetochore-microtubule attachments.

**Figure supplement 1—source data 1.** Configuration and data files for simulations used in *Figure 3—figure supplement 1*.

---

simultaneous biorientation increases with the angular stiffness of the third binding site. The second value of the angular potential for progressive restriction was less important (*Figure 3—figure supplement 1A*): varying it did not significantly change the fraction of simultaneous biorientation.

Because of the importance of progressive restriction in our model, we additionally examined whether varying the number of allowed kinetochore-MT attachments might affect how easily biorientation is achieved, but found no significant effect (*Figure 3—figure supplement 1B*). In these simulations, we chose how to vary the angular spring stiffness as the number of attachment sites varies. For fewer attachment sites, we chose the lower values of angular spring stiffnesses for progressive restriction that matched the reference stiffness. For increased number of attachments, the later attachments were fixed at an upper limit of 100 $k_BT$. In all cases, chromosome biorientation was not compromised.

## Error correction occurs through the destabilization of improper attachments

Progressive restriction of attachment reduces but does not eliminate erroneous kinetochore-MT attachments. Previous experimental work has shown that merotelic attachments are common in early mitosis and are corrected over time (*Cimini et al., 2003*) due to increased turnover of kinetochore MTs from the activity of Aurora B kinase (*DeLuca et al., 2006*; *Cimini et al., 2006*; *Gay et al., 2012*). To study this, we considered two different error correction models: biorientation-dependent stabilization and force-dependent stabilization. First, we implemented the rule-based model of misaligned destabilization by accelerating the detachment of kinetochore-MT attachments that are not amphitelic (*Figure 1F*). Because experimental work has demonstrated a decrease in kinetochore MT turnover by up to a factor of 65 in the presence of Aurora B inhibitors (*Cimini et al., 2006*), we varied the misaligned destabilization factor in the model, which quantifies the increased turnover of incorrect attachments, over a similar range from 1 to 100 (*Figure 3D*, *Video 4*).

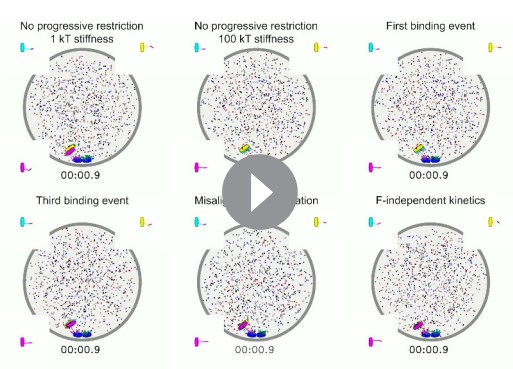

**Video 4.** Simulations of models with perturbation to kinetochore properties important for biorientation. Top left: Model lacking progressive restriction, with a common angular spring stiffnesses of 1 $k_BT$ for all attachments. A short bipolar spindle forms, but chromosomes are typically merotelically attached and do not biorient. Top middle: Model lacking progressive restriction, with a common angular spring stiffnesses of 100 $k_BT$ for all attachments. A long bipolar spindle forms, kinetochore-MT attachments are transient, and chromosomes do not generate significant inward force on the spindle. Top right: Model including progressive restriction with an angular spring stiffness of 20 $k_BT$ for the first binding event, leading to restricted attachments. A long bipolar spindle forms, and kinetochore-MT attachments are transient. Lower left: model including progressive restriction but with an angular spring stiffness of 20 $k_BT$ for the third binding event, leading to permissive attachments. Error correction is impaired, and chromosomes are typically merotelically attached. Lower middle: Model lacking misaligned destabilization. Error correction is impaired. Lower right: Model with force-independent attachment kinetics. Kinetochore-MT attachments are not stabilized under tension from depolymerizing microtubules, leading to short-lived biorientation.

https://elifesciences.org/articles/48787#video4

---

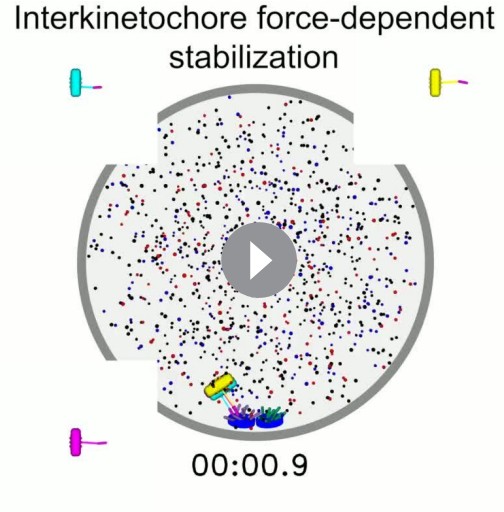

**Video 5.** Simulation of a model with interkinetochore force-dependent attachments. The spindle forms in a few minutes, and chromosomes form stable, bioriented attachments. Zoomed views of chromosomes shows them forming load-bearing attachments to the tips of MTs. The interkinetochore characteristic force is 1.67 pN.

https://elifesciences.org/articles/48787#video5

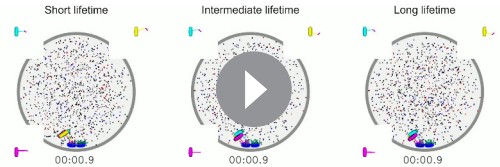

**Video 6.** Simulations of models with varying kinetochore-MT attachment lifetime. Left: Model with short attachment lifetime in which the kinetochore-MT binding and unbinding rates are 4 times larger than in the reference model. Biorientation is somewhat compromised. Middle: Model with intermediate attachment lifetime in which the kinetochore-MT binding and unbinding rates are 2 times larger than in the reference model. Right: Model with long attachment lifetime in which the kinetochore-MT binding and unbinding rates are 2 times smaller than in the reference model. Biorientation is preserved and the spindle undergoes large length fluctuations.

https://elifesciences.org/articles/48787#video6

Consistent with experimental results, biorientation is nearly eliminated in the absence of misaligned destabilization. Biorientation time in the model is maximum when the misaligned destabilization factor is 70, comparable to the experimental value. This demonstrates the importance of error correction in the model.

The biorientation-dependent model has the disadvantage that it cannot test any mechanisms by which incorrect attachments are destabilized. We therefore additionally tested a force-dependent error correction model, based on previous results that kinetochore-MT attachments are stabilized by force (*Nicklas and Koch, 1969*; *Cane et al., 2013*). We modeled the kinetics of kinetochore-MT attachments as a function of interkinetochore tension, with the rates decreasing with force (*Figure 3E*, *Video 5*), controlled by a a characteristic force for significant stabilization.

The force-stabilization model of error correction that we implemented experiences the initial problem of biorientation (IPBO): a bioriented attachment that has just formed is not under tension, and therefore is not stable (*Zhang et al., 2013*; *Kalantzaki et al., 2015*; *Tubman et al., 2017*). Consistent with this, we found implementing force-dependent stabilization alone did not lead to biorientation. Recent work has suggested that the IPBO may be solved by initial syntelic-like attachments that are end-on between the kinetochore face near a pole, and lateral to the kinetochore farther from that same pole (*Kuhn and Dumont, 2017*). Therefore, we varied parameters in the model that might facilitate tension generation before biorientation, including the angular spring constants of the interkinetochore spring, the characteristic angular factor for binding high angles to the

**Table 6.** Force-dependent error correction model parameters.

| Parameter | Symbol | Value | Notes |
|---|---|---|---|
| Inter-kinetochore stabilization force | $F_{EC,0}$ | 1.67 pN | Optimized |
| Rotational spring constant | $\kappa_{C,u}$ | 925 pN nm rad$^{-1}$ | Optimized |
| Alignment spring constant | $\kappa_{C,v}$ | 925 pN nm rad$^{-1}$ | Optimized |
| Angular characteristic factor | $\chi_c$ | 0.08 | Optimized |
| Side concentration | $c_{AF,side}$ | 0.32 nm$^{-1}$ | Optimized |
| Kinesin-5 number | $N_{K5}$ | 200 | Optimized |

kinetochore plate, the effective concentration for binding laterally, and the number of kinesin-5 motors, which affect overall spindle force generation. We were able to achieve long-lived biorientation in the force-dependent error correction model with model parameters that favored end-on over lateral attachments, inhibited attachments at high angle, and allowed sister kinetochores to more easily reorient (*Table 6*).

In this version of the model, we then varied the characteristic force that controls how much attachments are stabilized by force (*Figure 3E*, *Video 5*). The characteristic force is the value of the interkinetochore force at which attachments are stabilized by a factor of two, so a small value reflects rapid variation of attachment stability with force, while an infinite value means that attachments are force independent. We found that the model is sensitive to the value of this characteristic force, with best performance of the model at a characteristic force of 1.67 pN. Higher or lower values decrease cumulative biorientation by up to a factor of two.

## Persistent biorientation is achieved through force-dependent kinetochore-microtubule attachment

Once amphitelic kinetochore-MT attachments are formed, they must be maintained for biorientation to persist. Attachments between single MTs and purified budding-yeast kinetochores were altered by force applied to the kinetochore, even in the absence of Aurora kinase (*Akiyoshi et al., 2010*; *Miller et al., 2016*). In particular, the kinetochore-MT attachment lifetime increased with tension when kinetochores were attached to depolymerizing MTs, an effect dependent on a TOG protein (*Akiyoshi et al., 2010*; *Miller et al., 2016*). Consistent with this, we implemented force dependence of attachments in the model (*Figure 1G*). This effect is required to maintain biorientation: if we eliminate the force dependence of attachment kinetics, biorientation is nearly abolished in the model (*Figure 3F*, *Video 4*). To understand which force-dependent rate is most important for this effect, we added them back to the model one at a time. The increase in attachment lifetime of a kinetochore bound to a shrinking MT is the key force-dependent rate, because making this the only force-dependent lifetime in the model restores nearly all biorientation compared to the model with all rates force-dependent (*Figure 3F*). This demonstrates that maintenance of biorientation requires kinetochore-MT attachments to persist during MT depolymerization.

## Slow turnover of kinetochore-microtubule attachments can cause spindle length fluctuations

Spindle length regulation (*Dumont and Mitchison, 2009*; *Goshima and Scholey, 2010*; *Syrovatkina et al., 2013*; *Hepperla et al., 2014*; *Nannas et al., 2014*; *Rizk et al., 2014*) can be understood using the force-balance model of Saunders and Hoyt in which plus-end directed sliding motors produce outward force, and minus-end directed sliding motors and chromosomes produce inward force (*Saunders and Hoyt, 1992*; *Nabeshima et al., 1998*; *Goshima et al., 1999*; *Severin et al., 2001*; *Tolić-Nørrelykke et al., 2004*; *Bouck and Bloom, 2007*; *Stephens et al., 2013*; *Syrovatkina et al., 2013*; *Costa et al., 2014*; *Zheng et al., 2014*; *van Heesbeen et al., 2014*; *Syrovatkina and Tran, 2015*). The force-balance model has been used in mathematical models of spindles in yeast (*Gardner et al., 2005*; *Gardner et al., 2008*; *Chacón et al., 2014*; *Hepperla et al., 2014*; *Ward et al., 2015*; *Blackwell et al., 2017a*; *Rincon et al., 2017*; *Lamson et al., 2019*), and *Drosophila* (*Cytrynbaum et al., 2003*; *Cytrynbaum et al., 2005*; *Wollman et al., 2008*; *Civelekoglu-Scholey and Scholey, 2010*) cells. This work has focused on spindle length at steady state, not dynamic changes. However, some fission-yeast mutants exhibit large fluctuations in spindle length in which the bipolar spindle assembles, but then shortens or falls apart, known as spindle collapse (*Bratman and Chang, 2007*; *Griffiths et al., 2008*; *Choi et al., 2009*; *Hsu and Toda, 2011*; *Masuda et al., 2013*; *Wälde and King, 2014*; *Syrovatkina et al., 2013*; *Gergely et al., 2016*). Remarkably, fission-yeast double mutants can have wild-type average metaphase spindle length, but much larger fluctuations than wild-type (*Syrovatkina et al., 2013*). The underlying mechanisms of large spindle length fluctuations have remained unclear, in part because apparently contradictory changes can cause it. For example, deletion of proteins known either to stabilize (*Bratman and Chang, 2007*) or destabilize MTs (*Gergely et al., 2016*) can both lead to large spindle length fluctuations. In recent work we examined how deletion of the kinesin-8 motor proteins could contribute to large spindle length fluctuations in fission yeast (*Gergely et al.,*

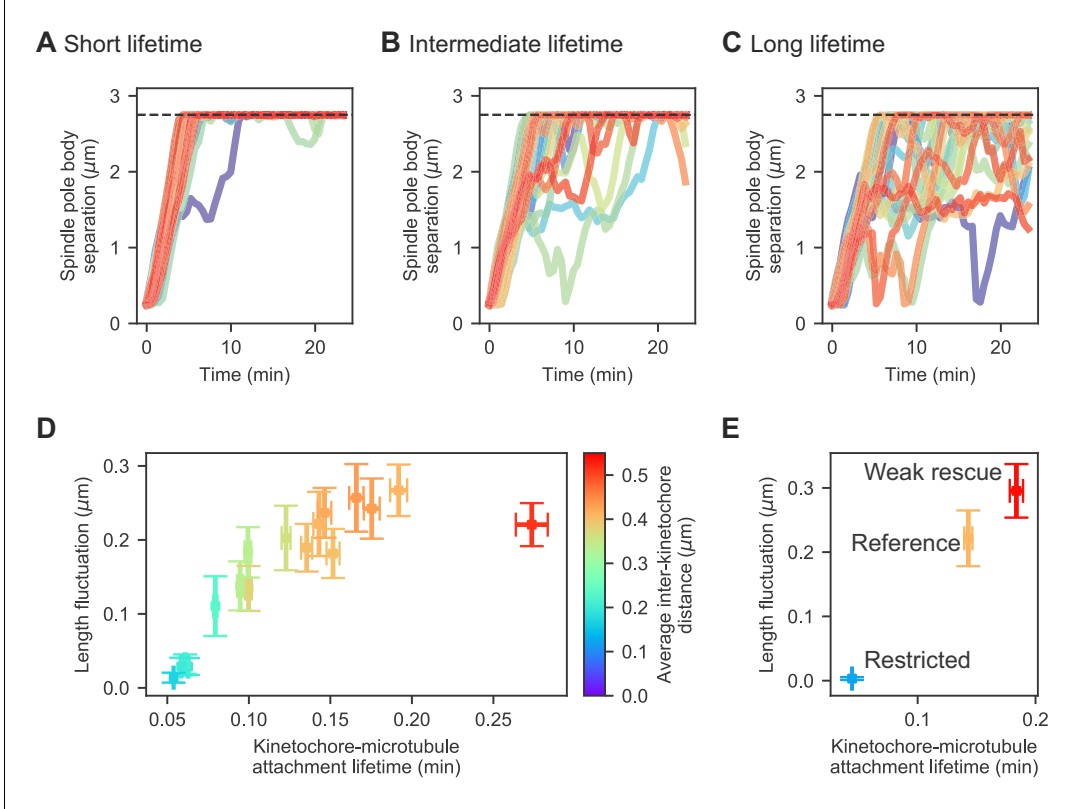

**Figure 4.** Changes in kinetochore-MT attachment turnover alter spindle length fluctuations. (**A–C**) Spindle length versus time for 24 simulations of the same model, with (**A**) short (1/4 the reference value), (**B**) intermediate (1/2 the reference value), and (**C**) long (twice the reference value) kinetochore-MT attachment lifetime. (**D**) Length fluctuation magnitude versus measured kinetochore-MT attachment lifetime and average interkinetochore stretch (color) for bioplar spindles (corresponding to simulation time >10 min.). (**E**) Length fluctuation magnitude versus measured kinetochore-MT attachment lifetime and average interkinetochore stretch (color) for the reference, restricted, and weak rescue models (N = 24 simulations per data point).

The online version of this article includes the following source data for figure 4:

**Source data 1.** Configuration and data files for simulations used in *Figure 4*.

---

*2016*), but a general understanding of this phenomenon is lacking. Therefore, we sought to understand what mechanisms might lead to large length fluctuations.

One key determinant of the magnitude of spindle length fluctuations is the lifetime of kinetochore-MT attachments (*Figure 4*, *Video 6*). We quantified the magnitude of length fluctuations by determining the standard deviation in spindle length over time after spindle elongation for each individual simulation of the model, then averaging that standard deviation over multiple model runs with the same parameters. This measure of length fluctuations increases with kinetochore-MT attachment lifetime: the longer the lifetime, the larger the fluctuations (*Figure 4A–D*).

To understand this result, note that for long-lived attachment, the force exerted by a stretched kinetochore can grow over time to a larger value: long-lived attachment allows multiple MTs to bind per kinetochore, exert greater force, and stretch apart the sisters. This allows larger inward force to be exerted on the spindle by attached kinetochores. Indeed, the average interkinetochore distance increases with kinetochore-MT attachment lifetime (*Figure 4D*). Thus, slow cycles of attachment and detachment lead to slowly varying force on the spindle that causes its length to fluctuate. In the opposite limit, short-lived kinetochore-MT attachment causes relatively quick turnover, limiting interkinetochore stretch, inward force, and variation in inward force.

Alteration in kinetochore-MT attachment lifetime could occur through multiple molecular mechanisms. To illustrate how this could occur, we considered two perturbations to the model that have downstream effects on both lifetime and length fluctuations (*Figure 4E*). The first perturbation is a restricted attachment model, in which the angular spring constant of attachment discussed above (*Figure 3A*) is set to 100 $\mathrm{k_B T}$ for all attachments. In this case, attachments rarely form and when

**Video 7.** Simulations of reference, restricted, and weak rescue models. Left: The reference model shows typical spindle length fluctuations. Middle: The restricted attachment model shows minimal length fluctuations, because transient kinetochore-MT attachments lead to low inward force on the spindle from chromosomes. Right: The weak rescue model shows large spindle length fluctuations, because kinetochore MTs remain attached while depolymerizing, leading to high and fluctuating inward force on the spindle from chromosomes.

https://elifesciences.org/articles/48787#video7

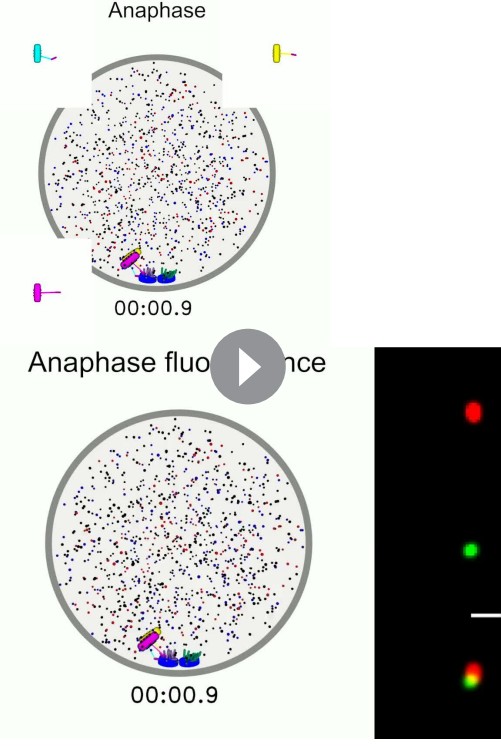

**Video 8.** Simulations of anaphase chromosome segregation. Top: Simulation video showing that separation of the sister chromatids occurs after 4.45 min of the simultaneous biorientation of all three chromosomes. The zoomed views show the chromosomes achieving biorientation before segregating to the spindle poles. Lower: Simulation video (left) and simulated fluorescence microscopy images (right), with red MTs and green kinetochore (scale bar 1 μm). The simulated fluorescence images are rotated so that the spindle is vertical. Anaphase occurs at 7:09.

https://elifesciences.org/articles/48787#video8

formed, their lifetime is short (<0.05 min on average). As a result, the force produced by interkinetochore stretch is small and does not vary much, leading to small length fluctuations in the model (<0.01 μm on average). The opposite limit can occur in a model in which the force-dependent rescue of kinetochore MTs is greatly reduced, by increasing the force constant from 6.4 pN to 12.8 pN (this reduces the force sensitivity of rescue, see Appendix 1). This causes kinetochore MTs to depolymerize for longer time, and because kinetochore-MT attachments are stabilized during depolymerization, this change dramatically increases the attachment lifetime to 0.2 min. As a result, interkinetochore stretch can increase, and length fluctuations correspondingly increase (0.3 μm).

This analysis suggests that altered kinetochore-MT attachment lifetime could be a downstream effect that may result from the diverse mutations observed to cause spindle length fluctuations in *S. pombe*. We note that the effect of lifetime may not be the only source of spindle length fluctuations: other mutations that lead to slow changes in force exerted on the spindle could have similar effects.

## Force generation in the spindle varies during spindle elongation

The force-balance model can explain why multiple perturbations alter steady-state spindle length, including mutation of motors and microtubule-associated proteins (*Syrovatkina et al., 2013*; *Hepperla et al., 2014*), and chromosome/kinetochore number and chromatin stiffness (*Chacón et al., 2014*; *Nannas et al., 2014*). However, it can be challenging to distinguish direct from indirect effects of altering force balance. For example, the force-balance model posits that minus-end-directed kinesin-14 motors contribute inward force that shortens the spindle, so their deletion would be expected to lead to longer spindles. However, in budding yeast, kinesin-14 deletion instead leads to shorter spindles, because kinesin-14 helps bundle spindle MTs, allowing kinesin-5 motors to generate greater outward force when kinesin-14 is present (*Hepperla et al., 2014*). Similarly, kinesin-8 deletion in fission yeast leads to longer spindles, but this is likely due to effects of

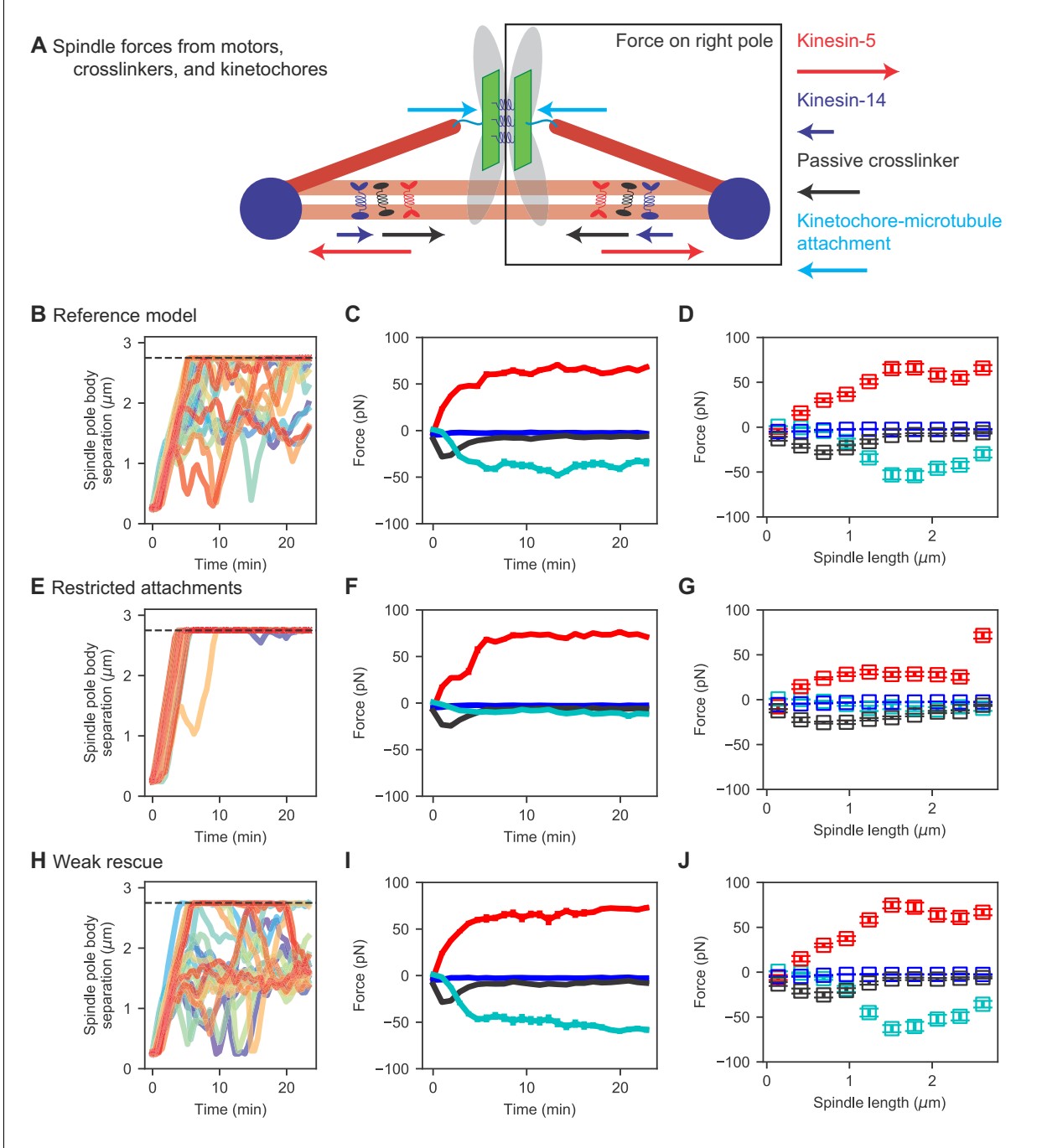

**Figure 5.** Spindle force generation varies as the spindle assembles and elongates. (**A**) Schematic of force generation along the spindle axis, showing kinesin-5 motors exerting outward force (red) and kinesin-14 (dark blue), crosslinkers (black), and kinetochore-MT attachment to stretched chromosomes (light blue) exerting inward force. (**B, E, H**) Spindle length versus time, (**C, F, I**) average spindle axis force versus time, and (**D, G, J**) average spindle axis force versus spindle length for three different models: (**B–D**) the reference model, (**E–G**) the restricted attachment model, and (**H–J**) the weak rescue model (N = 24 simulations per data point).

The online version of this article includes the following source data for figure 5:

**Source data 1.** Configuration and data files for simulations used in *Figure 5*.

**Table 7.** Anaphase parameters.

| Anaphase | Symbol | Value | Notes |
|---|---|---|---|
| Integrated simultaneous biorientation time | $\tau_{SAC}$ | 4.45 min | Chosen |
| Anaphase attachment rate | $k_{AF,anaphase}$ | 0.00007 s$^{-1}$ | Chosen |
| Anaphase MT depoly speed | $v_{anaphase,s,0}$ | 2.2 μm min$^{-1}$ | Chosen |

this motor on MT dynamics rather than direct inward force generation by kinesin-8 (*Syrovatkina et al., 2013*; *Gergely et al., 2016*).

To better understand direct and indirect changes in spindle length, we examined the force produced by spindle molecules as the spindle elongates, averaged over many simulation runs (*Figure 5*, *Video 7*). In this analysis, we considered each half-spindle separately, and calculated the total force exerted along the spindle axis produced by separate force-generating elements: outward force by kinesin-5 motors, and inward force by kinesin-14 motors, passive crosslinkers, and kinetochore-MT attachments (*Figure 5A*). We computed spindle length as a function of time (*Figure 5B,E,H*), force as a function of time (*Figure 5C,F,I*) and spindle length (*Figure 5D,G,J*) in the reference, restricted attachment, and weak rescue models.

## The early bipolar spindle forms due to motors and crosslinkers, not chromosomes

Force generation by kinesin-5 motors, kinesin-14 motors, crosslinkers, and chromosomes changes significantly as the bipolar spindle assembles. For early time (up to 5 min) when spindles are short (up to 1 μm), motors and crosslinkers exert force that slowly increases in magnitude up to a few tens of pN, but chromosomes exert almost no force (*Figure 5C,F,I*, *Video 7*). Because chromosomes are not bioriented on the spindle during initial SPB separation, they do not exert significant inward force. This result is consistent with our previous work, which demonstrated that initial bipolar spindle assembly can occur in a model lacking chromosomes (*Blackwell et al., 2017a*; *Rincon et al., 2017*; *Lamson et al., 2019*).

The outward sliding force produced by kinesin-5 motors increases approximately linearly with spindle length, as the length of antiparallel MT overlaps increases during spindle assembly (*Figure 5D,G,J*). This agrees with the experimental result that the sliding force generated by kinesin-5 motors is proportional to overlap length (*Shimamoto et al., 2015*). The inward force generated by kinesin-14 motors is small, as in previous work that has shown that kinesin-14 is less effective at force generation that kinesin-5 (*Hentrich and Surrey, 2010*) and that in the spindle kinesin-14 may be more important to align spindle MTs than to generate force directly (*Hepperla et al., 2014*).

During initial spindle assembly, crosslinkers play the primary role of maintaining antiparallel MT overlaps in opposition to the sliding activity of kinesin-5. Remarkably, we find that the inward force generated by passive crosslinkers initially increases with spindle length to approximately 25 pN when the spindle is 0.75 μm long. Beyond this point, the crosslinker force steadily decreases, dropping to near zero within a few minutes (*Figure 5C,F,I*). This is consistent with previous results on force generation by the crosslinker Ase1, which found large force for small overlaps that drops significantly as overlaps become larger (*Lansky et al., 2015*). Therefore, our results support a picture of early spindle assembly in which high braking force by crosslinkers on short antiparallel MT overlaps oppose the outward force generated by kinesin-5. This highlights the key role of crosslinkers in early spindle assembly suggested previously (*Blackwell et al., 2017a*; *Rincon et al., 2017*; *Lamson et al., 2019*).

## Metaphase spindle length is determined primarily by interkinetochore stretch and kinesin-5 motors

Once the spindle elongates sufficiently to separate SPBs by 1 μm, there is a transition in the primary contributer to spindle force. In this regime, chromosomes biorient and the inward force from interkinetochore stretch becomes significant, balancing outward force from kinesin-5 motors (*Figure 5C,F, I*). This balance is crucial to setting metaphase spindle length.

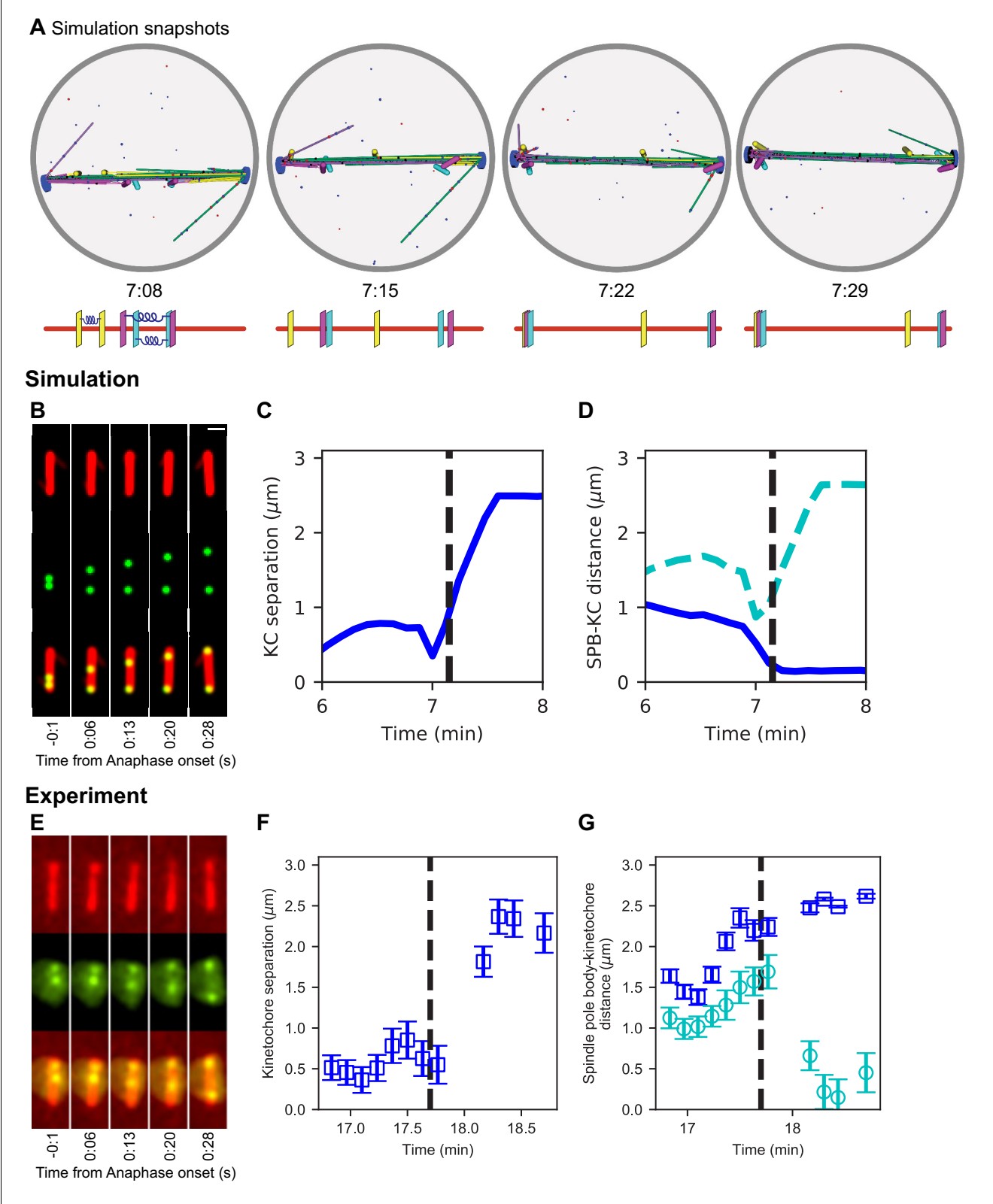

**Figure 6.** Chromosome segregation in the model and comparison to experiments. (A) Image sequence of simulation of chromosome segregation after anaphase is triggered, rendered from a three-dimensional simulation. Anaphase begins immediately after the first image. Lower, schematic showing kinetochore position along the spindle. Time shown in minutes:seconds. (B–D) Simulation results. (B) Simulated fluorescence microscopy images with MTs (red) and a single kinetochore pair (green). Time shown in minutes:seconds. (C) Spindle pole body-kinetochore distance, and (D) interkinetochore

*Figure 6 continued on next page*

*Figure 6 continued*

distance versus time from the simulation shown in (**B**), sampled at a rate comparable to the experimental data in (**E–G**). (**E–G**) Experimental results. Maximum-intensity projected smoothed images from time-lapse confocal fluorescence microscopy of fission yeast with *mCherry-atb2* labeling MTs (red) and *cen2-GFP* labeling the centromere of chromosome 2 (green). Time shown in minutes:seconds. (**E**) Spindle length, (**F**) spindle pole body-kinetochore distance, and (**G**) interkinetochore distance versus time from the experiment shown in (**E**).

The online version of this article includes the following source data for figure 6:

**Source data 1.** Configuration and data files for simulations used in *Figure 6*.

To perturb this force balance, we considered two additional models discussed above (*Figure 4E*) with restricted attachment and weak rescue. When attachment is restricted, chromosomes rarely biorient and the inward force from chromosomes is small for spindles of all length. This leads to unbalanced force from kinesin-5 motors and long spindles (*Figure 5E–G*, *Video 7*). When MT rescue is reduced, interkinetochore stretch is larger and the inward force from stretched sister kinetochores increases (*Figure 5H–J*, *Video 7*). This leads to shorter metaphase spindle length and a corresponding increase in force from stretched kinetochores.

## Chromosome segregation can occur via the same mechanisms that assemble the spindle

After developing the model of spindle assembly and chromosome biorientation, we examined what additional mechanisms were required for the model to segregate chromosomes to the poles. Relatively few changes are required for robust chromosome segregation, suggesting that significant new mechanisms are not required in anaphase for chromosome segregation. The rules added to the model for anaphase A include severing the chromatin spring between kinetochores (based on cumulative time the chromosomes are bioriented), stabilization of kinetochore-MT attachment, and depolymerization of MTs (*Table 7*). With these additions to the model, simulations consistently segregate chromosomes to the poles (*Figure 6A–D*, *Video 8*). We compared our simulations to experimental measurements of chromosome segregation, and found similar speed of chromosome movement to the poles and separation of sisters (*Figure 6E–G*), as expected from the choice of MT depolymerization speed in the anaphase model.

## Discussion

The computational model of mitosis presented here can biorient chromosomes as the spindle assembles. This framework allows us to examine which functions are most important to assemble a bipolar spindle, attach kinetochores to spindle MTs, biorient chromosomes, and segregate them to the poles (*Figure 1*; *Video 1*). Our model was refined with experimental data on spindle structure, spindle elongation, and chromosome movements in fission yeast, leading to quantitative agreement with the data (*Figure 2*; *Video 2*). The reference model results match previous genetics that found that kinesin-5 motors and CLASP are essential for bipolar spindle assembly (*Hagan and Yanagida, 1990*; *Hagan and Yanagida, 1992*; *Bratman and Chang, 2007*; *Blackwell et al., 2017a*), which suggests that the model captures key features needed to provide insight into mitotic mechanism.

Three ingredients are required for long-lived biorientation in the model (*Figure 3*; *Video 4*). Kinetochores shield themselves from merotely by progressive restriction of attachment. Inclusion of this effect in the model was motivated by recent work on the monopolin complex in fission yeast (*Gregan et al., 2007*) and attachment-driven compaction of mammalian kinetochores (*Magidson et al., 2015*). Progressive restriction has two key effects: it promotes proper attachment by favoring binding of microtubules from the same pole that is already attached to the kinetochore, and simultaneously creates a torque that helps to reorient the kinetochore on the spindle. In previous work, the monopolin complex components Pcs1/Mde4 were found not to be essential in fission yeast (*Gregan et al., 2007*), but in our model completely removing progressive restriction abolishes biorientation (*Figure 3*). This suggests the possibility that in fission yeast, other molecules may contribute to progressive restriction in addition to monopolin.

Mimicking the effects of Aurora B kinase by including destabilization of misaligned attachments allows the model to achieve robust error correction. Destabilization by approximately a factor of 70 gives the highest degree of biorientation the model. This is similar to the degree of destabilization

previously estimated to occur due to Aurora B (*Cimini et al., 2006*), further suggesting that the model produces biologically relevant results.

To maintain long-lived biorientation in the model, kinetochore-MT attachment lifetime must increase with tension during microtubule depolymerization. This catch-bond behavior has been previously measured for purified budding-yeast kinetochores attached to single microtubules (*Akiyoshi et al., 2010*; *Miller et al., 2016*). Without this force dependence, kinetochores frequently detach from depolymerizing MTs and lose biorientation. Our model achieves biorientation for the longest time with an increased force-sensitivity of attachment compared to experimental measurements, a difference that would be of interest to explore in future work.

The timing of spindle assembly and biorientation in the model were consistent with those quantified experimentally. A current difference between the model and experiment is that we find ongoing turnover of kinetochore-MT attachments, so that biorientation can be lost once established. This is in contrast to previous experimental work, which suggests that for metaphase spindles, once biorientation is established it is rarely lost (*Waters et al., 1996*; *Nicklas, 1997*; *Yoo et al., 2018*). The mechanisms underlying this difference are an open question.

Using our model, we studied the origins of large spindle length fluctuations (*Figure 4*; *Video 6*). While previous work has examined regulation of spindle length (*Syrovatkina et al., 2013*; *Hepperla et al., 2014*; *Nannas et al., 2014*; *Rizk et al., 2014*), what mechanisms might drive large fluctuations in spindle length over time have been less well-studied. We identified the lifetime of kinetochore-MT attachment as a determinant of the degree of spindle length fluctuations. Long attachment lifetime allows bioriented chromosomes to become more stretched, leading to large, slowly varying inward force on the spindle. Our results suggest why large spindle length fluctuations have not been seen in larger spindles in vertebrate cells: in *S. pombe*, a relatively small number of kinetochores and MTs contribute to spindle length, and therefore the changing force on the three chromosomes can have a significant effect on the spindle. In vertebrate spindles with tens of thousands of MTs, changes in force on a small number of kinetochores contribute only a small fractional change to overall force on the spindle, leading to smaller fluctuations.

To understand how force generation changes as the spindle assembles, we quantified the force generated by different classes of spindle molecule (*Figure 5*; *Video 7*). The early spindle has almost no force generation from interkinetochore stretch because chromosomes are rarely bioriented at this stage. Instead, the early spindle is characterized by outward force from kinesin-5 motors that is resisted by crosslinkers. Consistent with earlier work (*Lansky et al., 2015*), the force from crosslinkers is highest when MT antiparallel overlaps are short and drops as the spindle elongates. Once the bipolar spindle is formed and chromosomes are bioriented, attached chromosomes provide significant inward force that opposes the outward force of kinesin-5 motors. These results suggest that the many mutations that alter spindle length in fission yeast (*Syrovatkina et al., 2013*) might act indirectly by altering kinesin-5 force generation or interkinetochore stretch.

Remarkably, the model is able to transition to anaphase A and robustly segregate chromosomes to the poles with a small number of additional rules (*Figure 6*; *Video 8*). Overall, our work provides a powerful framework for testing spindle assembly mechanisms that can inform future experimental studies.

## Acknowledgements

We thank Jeffrey K Moore for useful discussions. This work was funded by NSF grants DMR-1725065 (MDB), DMS-1620003 (MAG and MDB), and DMR-1420736 (MAG); NIH grants K25GM110486 (MDB), R01GM124371 (MDB); a fellowship provided by matching funds from the NIH/University of Colorado Biophysics Training Program (AL); and use of the Summit supercomputer, supported by NSF grants ACI-1532235 and ACI-1532236.

## Additional information

### Funding

| Funder | Grant reference number | Author |
| --- | --- | --- |
| National Science Foundation | DMR-1725065 | Meredith D Betterton |

| National Science Foundation | DMS-1620003 | Matthew A Glaser<br>Meredith D Betterton |
|---|---|---|
| National Science Foundation | DMR-1420736 | Matthew A Glaser |
| National Institutes of Health | K25GM110486 | Meredith D Betterton |
| National Institutes of Health | R01GM124371 | Meredith D Betterton |
| University of Colorado | Biophysics Training Program Fellowship | Adam R Lamson |
| National Science Foundation | ACI-1532235 | Christopher Edelmaier<br>Adam R Lamson<br>Zachary R Gergely<br>Saad Ansari<br>Robert Blackwell<br>J Richard McIntosh<br>Matthew A Glaser<br>Meredith D Betterton |
| National Science Foundation | ACI-153223 | Christopher Edelmaier<br>Adam R Lamson<br>Zachary R Gergely<br>Saad Ansari<br>Robert Blackwell<br>J Richard McIntosh<br>Matthew A Glaser<br>Meredith D Betterton |
| National Institutes of Health | | Adam R Lamson |

The funders had no role in study design, data collection and interpretation, or the decision to submit the work for publication.

## Author contributions

Christopher Edelmaier, Conceptualization, Data curation, Software, Formal analysis, Validation, Investigation, Visualization, Methodology, Writing - original draft, Writing - review and editing; Adam R Lamson, Data curation, Software, Formal analysis, Validation, Visualization, Methodology; Zachary R Gergely, Investigation; Saad Ansari, Formal analysis; Robert Blackwell, Conceptualization, Software; J Richard McIntosh, Conceptualization; Matthew A Glaser, Conceptualization, Supervision, Funding acquisition, Methodology; Meredith D Betterton, Conceptualization, Resources, Supervision, Funding acquisition, Methodology, Writing - original draft, Project administration, Writing - review and editing

## Author ORCIDs

Christopher Edelmaier https://orcid.org/0000-0002-3673-1310
Meredith D Betterton https://orcid.org/0000-0002-5430-5518

## Decision letter and Author response

Decision letter https://doi.org/10.7554/eLife.48787.sa1
Author response https://doi.org/10.7554/eLife.48787.sa2

# Additional files

## Supplementary files

• Source code 1. Code for simulation and analysis framework for confined SPB simulations. Requires C++ compiler, and GSL, GLEW, python2, python3, libyaml, FFTW, GLFW, xQuartz, freeGLUT, libpng, ffmpeg, pkg-config, and png++ libraries. Python libraries should include matplotlib, numpy, opencv-python, panda3d, pandas, PyYAML, and scipy for analysis framework. Used for all simulations except *Figure 2—figure supplement 1*, panel D: Soft nuclear envelope. Untar and unzip SourceCodeFile1.tar.gz, then use the accompanying Makefile and MakefileIncmk to compile on your system.

• Source code 2. Code for simulation and analysis framework for free SPB simulations. Requires C++ compiler, and armadillo, GSL, GLEW, python2, python3, libyaml, FFTW, GLFW, xQuartz, freeGLUT, libpng, ffmpeg, pkg-config, and png++ libraries. Python libraries should include matplotlib, numpy, opencv-python, panda3d, pandas, PyYAML, and scipy for analysis framework. Used only for *Figure 2—figure supplement 1*, panel D: Soft nuclear envelope. Untar and unzip SourceCodeFile2.tar.gz, then use the accompanying Makefile and MakefileIncmk to compile on your system.

• Transparent reporting form

## Data availability

All data generated or analysed during this study are included in the manuscript and supporting files.

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

## Appendix 1

### Extended methods

#### Computational model

Our group has developed a simulation framework for microtubule-motor active matter and mitotic spindle self-assembly (*Gergely et al., 2016*; *Blackwell et al., 2017b*; *Blackwell et al., 2017a*; *Rincon et al., 2017*; *Lamson et al., 2019*). The computational scheme alternates between Brownian dynamics (BD) and kinetic Monte Carlo (kMC) steps to evolve the system forward in time. BD describes how particles move in response to forces and torques in a highly viscous medium. KMC methods handle stochastic state transitions, such as binding and dynamic instability (*Blackwell et al., 2017a*).

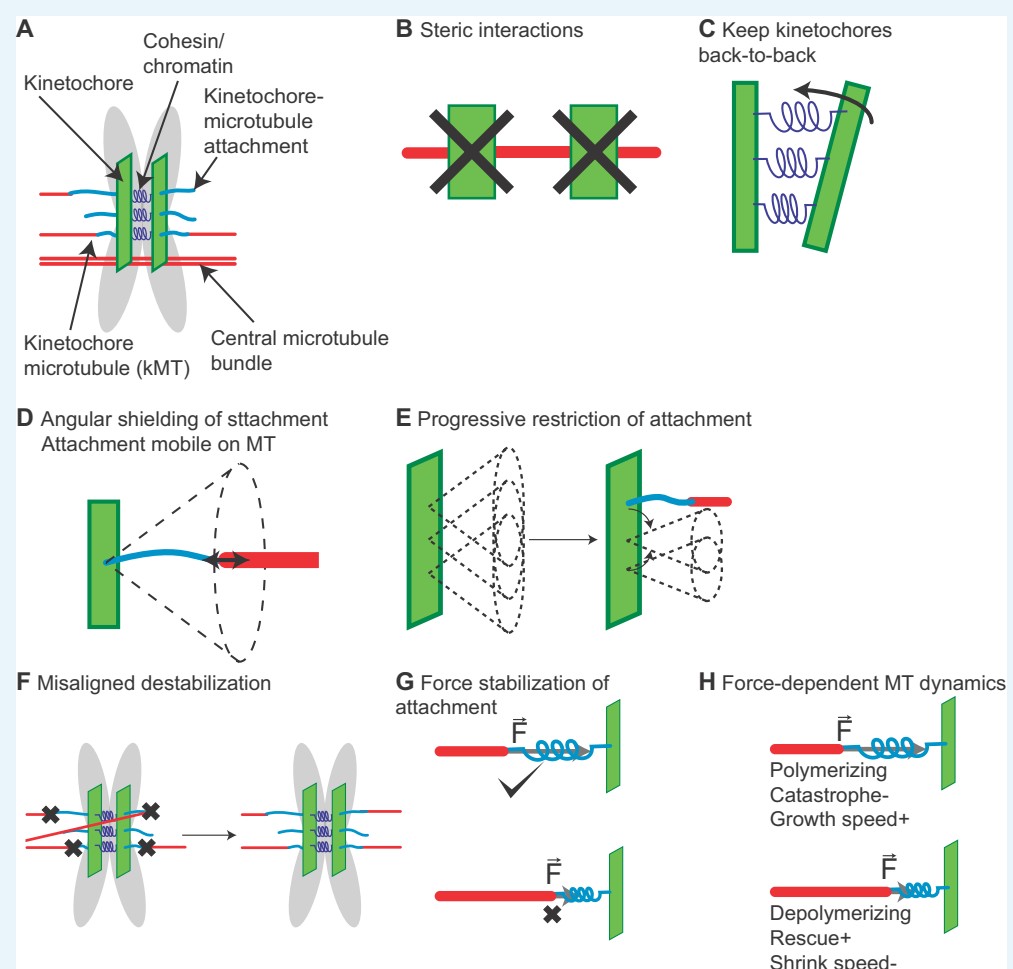

**Appendix 1—figure 1.** Chromosome model overview. (**A**) Chromosomes are modeled as sister chromatids and kinetochores held together by a cohesin/chromatin spring complex. Each kinetochore can attach up to three microtubules. (**B**) Steric interactions between MTs and kinetochores prevent overlap, while a soft steric repulsion exists between MTs and the centromeric DNA. (**C**) Kinetochores are kept back-to-back through a cohesin-chromatin spring complex that depends on relative kinetochore position and orientation. (**D**) The angular range of kinetochore-MT attachment is restricted based on the stiffness of an angular spring. (**E**) The angular restriction of kinetochore-MT attachment changes based on the number of bound MTs. (**F**) Attachments are destabilized when the chromosome is not properly bioriented. (**G**)

Attachment lifetime is force-dependent, with attachments to depolymerizing MTs under tension having longer lifetimes, while those to polymerizing MTs have their lifetime decreased under tension. (H) MT dynamics are force-dependent. Polymerizing MTs have increased growth speed and reduced catastrophe, while depolymerizing MTs have increased rescue and decreased shrinking speed.

### 1.1.1 Microtubules (MTs)

MTs are built of $\alpha$- and $\beta$-tubulin subunits that join end-to-end to form protofilaments. Approximately 13 side-by-side protofilaments form a hollow cylinder with distinct plus- and minus-ends. MTs undergo dynamic instability, in which they grow and shrink with speeds $v_g$ and $v_s$, transition from a shrinking state to a growing state (rescue) at rate $f_r$, and transition from growing to shrinking (catastrophe) at rate $f_c$(**Mitchison and Kirschner, 1984**). MT catastrophe rate increases with compressive force (**Janson et al., 2003**).

We model MTs as growing and shrinking spherocylinders that experience steric repulsion with other MTs and molecules. The typical MT length in fission yeast spindles (~1 μm) is much shorter than the MT persistence length (~1 mm), so we treat MTs inflexible filaments (**Tao et al., 2005**). Each MT has a center-of-mass coordinate $\mathbf{x}$, orientation $\mathbf{u}$, and length $L$ (**Blackwell et al., 2017a**). The MT position evolves according to

$$\mathbf{x}_i(t+\delta t) = \mathbf{x}_i(t) + \Gamma_i^{-1}(t) \cdot \mathbf{F}_i(t)\delta t + \delta\mathbf{x}_i(t), \tag{1}$$

where the random displacement $\delta\mathbf{x}_i(t)$ is Gaussian-distributed and anisotropic, with variance

$$\langle \delta\mathbf{x}_i(t)\delta\mathbf{x}_i(t)\rangle = 2k_BT\Gamma_i^{-1}(t)\delta t, \tag{2}$$

and $\Gamma_i^{-1}(t)$ is the inverse friction tensor

$$\Gamma_i^{-1}(t) = \gamma_\parallel^{-1}\mathbf{u}_i(t)\mathbf{u}_i(t) + \gamma_\perp^{-1}[\mathbf{I} - \mathbf{u}_i(t)\mathbf{u}_i(t)], \tag{3}$$

where $\gamma_\parallel$ and $\gamma_\perp$ are the parallel and perpendicular drag coefficients, and $\mathbf{F}_i(t)$ is the force on filament $i$ at time $t$. MT orientation evolves according to

$$\mathbf{u}_i(t+\delta t) = \mathbf{u}_i(t) + \frac{1}{\gamma_r}\mathbf{T}_i(t) \times \mathbf{u}_i(t)\delta t + \delta\mathbf{u}_i(t), \tag{4}$$

where $\gamma_r$ is the rotational drag coefficient, $\mathbf{T}_i(t)$ the torque, and $\delta\mathbf{u}_i(t)$ the random reorientation, which is Gaussian distributed with variance

$$\langle \delta\mathbf{u}_i(t)\delta\mathbf{u}_i(t)\rangle = 2k_BT/\gamma_r[\mathbf{I} - \mathbf{u}_i(t)\mathbf{u}_i(t)]\delta t, \tag{5}$$

where $\mathbf{I}$ is the identity matrix.

The drag coefficients $\gamma_\parallel$, $\gamma_\perp$, and $\gamma_r$ are recalculated at each time step based on the MT length $L$(**Blackwell et al., 2017a**; **Blackwell et al., 2017b**; **Blackwell et al., 2016**; **Gao et al., 2015b**; **Gao et al., 2015a**). Random translation and reorientation are treated in the body-frame of the MT. Random parallel displacements are

$$\delta x_\parallel = s_{\text{random}}\sqrt{\frac{2k_BT\delta t}{\gamma_\parallel}}R(t), \tag{6}$$

where $R(t)$ is a Gaussian random variate with $\sigma = 1.0$, and $s_{\text{random}}$ varies the strength of the random forces if necessary. Perpendicular displacements are

$$\delta x_\perp = s_{\text{random}}\sqrt{\frac{2k_BT\delta t}{\gamma_\perp}}R(t), \tag{7}$$

for each perpendicular dimension of the MT in the body-frame. Random reorientations are

$$\delta u = s_{\mathrm{random}} \sqrt{\frac{2k_B T \delta t}{\gamma_r}} R(t), \tag{8}$$

for each angle of the MT in the body-frame.

We model dynamic instability as a continuous stochastic process in which MTs in the polymerizing state grow with speed $v_g$, while those in the shrinking state depolymerize with speed $v_s$. MTs undergo catastrophe at rate $f_{c,0}$ and rescue at rate $f_{r,0}$. These rates are modified by interactions with crosslinkers, kinetochores, and the nuclear envelope. At each time step, kinetic Monte Carlo sampling is used to determine dynamic state transitions. Each MT stochastically switches between its states according to the dynamic instability parameters (**Table 1**; **Blackwell et al., 2017a**; **Kalinina et al., 2013**). Using previous methods (**Blackwell et al., 2017b**; **Blackwell et al., 2017a**; **Janson et al., 2003**; **Dogterom and Yurke, 1997**), force-induced catastrophe is implemented at MT plus-ends, according to an exponential force term $f_{\mathrm{cat}}(\mathbf{F}_\parallel) = \mathrm{f}_{\mathrm{cat},0} e^{\alpha_c \mathbf{F}_\parallel}$. Rather than explicitly modeling MT nucleation, we have chosen to have a fixed number of MTs with maximum and minimum length. When MTs reach the minimum length while undergoing catastrophe, they switch to the growing state. However, MTs that reach their maximum length pause, ensuring numerical stability for barrier interactions.

We model steric repulsion using the Weeks-Chandler-Anderson (WCA) potential

$$u_{\mathrm{wca}}(r_{\min}) = \begin{cases} 4k_B T \left[ \left(\frac{\sigma_{\mathrm{MT}}}{r_{\min}}\right)^{12} - \left(\frac{\sigma_{\mathrm{MT}}}{r_{\min}}\right)^6 \right] + k_B T, & r_{\min} < 2^{1/6}\sigma_{\mathrm{MT}} \\ 0, & r_{\min} \geq 2^{1/6}\sigma_{\mathrm{MT}}, \end{cases} \tag{9}$$

where $r_{\min}$ is the minimum distance between two finite line segments of length $l$ that defines the filament axes and $\sigma_{\mathrm{MT}}$ the effective rod diameter. Large forces are capped at a fixed value based on the size of the time step to prevent numerical instability (**Gao et al., 2015b**).

### 1.1.2 Nuclear envelope

The nuclear envelope is modeled as a shell of fixed radius $R$ centered at the origin. As for MT-MT interactions, MT-nuclear envelope interactions use the WCA potential

$$u_{\mathrm{wca,MT}}(r_{\min}) = \begin{cases} 4k_B T \left[ \left(\frac{\sigma_{MT}}{r_{\min}}\right)^{12} - \left(\frac{\sigma_{MT}}{r_{\min}}\right)^6 \right] + k_B T, & r_{\min} < 2^{1/6}\sigma_{MT} \\ 0, & r_{\min} \geq 2^{1/6}\sigma_{MT}, \end{cases} \tag{10}$$

where $r_{\min}$ is the minimum distance between the free end of the MT and the enclosing sphere with radius $R + \sigma_{MT}/2$. This allows for smooth continuation of the dynamics at the nuclear envelope, which has an effective radius of $R$. Similar to the MT-MT interactions, forces are capped to prevent instability for rare high-overlap events. As mentioned previously, the MT-nuclear envelope interaction enhances MT catastrophe (**Table 1**).

### 1.1.3 Spindle pole bodies

Spindle pole bodies (SPBs), the centrosomes of fission yeast, are embedded in the nuclear envelope during mitosis. MT minus-ends are tethered to the SPBs. We model SPBs as spherical caps confined to the surface of the nuclear envelope (**Blackwell et al., 2017a**). Each SPB has a right-handed coordinate system defined by $\hat{\mathbf{u}}$ which points inward from the SPB, and $\hat{\mathbf{v}}$ and $\hat{\mathbf{w}}$ which are arbitrary and perpendicular to one another. The equations of motion for an SPB constrained to move on the surface of the nuclear envelope are

$$\mathbf{u}_i(t+\delta t) = \mathbf{u}_i(t) - \frac{1}{R\gamma_t} \mathbf{F}_\parallel(t)\delta t + \delta \mathbf{u}_i(t), \tag{11}$$

where $\mathbf{F}_\parallel$ is the force in the plane tangent to the SPB and $\delta \mathbf{u}_i(t)$ is Gaussian-distributed with

variance $\langle \delta \mathbf{u}_i(t) \delta \mathbf{u}_i(t) \rangle = \frac{2k_B T}{R^2 \gamma_t}(\mathbf{I} - \mathbf{u}_i(t)\mathbf{u}_i(t))\delta t$. The corresponding rotational equation of motion for an SPB about its center is

$$\mathbf{v}_i(t+\delta t) = \mathbf{v}_i(t) + \frac{1}{\gamma_r}\mathbf{T}_{i,\text{body}}(t) \times \mathbf{v}_i(t)\delta t + \delta \mathbf{v}_i(t), \qquad (12)$$

where $\mathbf{T}_{i,\text{body}}$ is the torque on the SPB about the axis defined by $\mathbf{u}_i$.

The SPBs repel each other via the WCA potential

$$u_{\text{wca,SPB}}(\delta r_{\text{eff}}) = \begin{cases} 4k_B T\left[\left(\frac{\sigma_{\text{MT}}}{\delta r_{\text{eff}}}\right)^{12} - \left(\frac{\sigma_{\text{MT}}}{\delta r_{\text{eff}}}\right)^6\right] + k_B T, & \delta r_{\text{eff}} < 2^{1/6}\sigma_{\text{MT}} \\ 0, & \delta r_{\text{eff}} \geq 2^{1/6}\sigma_{\text{MT}}, \end{cases} \qquad (13)$$

where $\delta r_{\text{eff}} = \delta r - \sigma_{\text{SPB}} + \sigma_{\text{MT}}$.

Each SPB tethers the minus-ends of 14 MTs. Since the SPBs are three-dimensional rigid bodies confined to move on a two-dimensional surface, they have a fixed right-handed coordinate system that transforms according to the translation and rotation of the SPB. The attachment sites of the MT minus-ends are specified using this coordinate system. The tethers are modeled by a harmonic potential

$$u_{\text{teth}}(\mathbf{r}_{\text{MT,i}}, \mathbf{r}_{\text{teth,i}}) = \frac{1}{2}K_0\left(\left|\mathbf{r}_{\text{MT,i}} - \frac{L_i}{2}\hat{\mathbf{u}}_{\text{MT,i}} - \mathbf{r}_{\text{teth,i}}\right| - R_0\right)^2, \qquad (14)$$

where $L_i$ is the length of MT $i$, $\mathbf{r}_{\text{MT,i}}$ and $\hat{\mathbf{u}}_{\text{MT,i}}$ are the center of mass position and unit orientation vector for MT $i$ respectively, and $\mathbf{r}_{\text{teth,i}}$ is the vector connecting MT $i$'s tether position on the spindle pole body to the minus end of MT $i$. Torques on the MT are calculated using the force applied to the minus end of the MT associated with tether $i$. The tether springs do not interact with one another or any other objects in the system other than through the tethering potential (**Table 1**).

### 1.1.4 Soft nuclear envelope

In our model, SPBs are confined to move on a spherical shell of radius $R$, and MTs experience a steric interaction with this spherical shell. This limits the physical realism of the model, because it neglects the ability of the nuclear envelope to deform under force. The rigid nuclear envelope could lead to situations where the force on the spindle from the nuclear envelope sets the spindle length, rather than allowing spindle length to be determined by force balance between the nuclear envelope, motor and crosslinker proteins, and chromosomes. In order to address this issue, we have implemented changes to more realistically model the interactions between MTs, SPBs, and the nuclear envelope.

In the soft nuclear envelope model, SPBs are no longer confined to move on the spherical shell of the nuclear envelope. Instead, SPBs can freely translate and rotate in three dimensions. For the SPBs we implemented previously developed algorithms for 3D translational and rotational movement of rigid Brownian objects (**Ilie et al., 2015**). In this model, each SPB is defined by its center of mass coordinates $\mathbf{r}_i(t)$ and a quaternion describing its orientation $\mathbf{q}_i(t)$. This quaternion allows for the exact description of the unit coordinate axes that lie on the surface of the SPB ($\mathbf{u}$, $\mathbf{v}$, and $\mathbf{w}$). Translational motion for each SPB is described by the equation

$$r_\alpha(t+\Delta t) - r_\alpha(t) = A_{\alpha\gamma}\mu_{\gamma\delta}^{tb}A_{\beta\delta}F_\beta\Delta t + A_{\alpha\gamma}(\sqrt{\mu^{tb}})_{\gamma\beta}\Theta_\beta^t\sqrt{2k_B T\Delta t}, \qquad (15)$$

where $\mathbf{A}$ is the current rotation matrix describing the orientation of the SPB expressed in its homogeneous form, $\mu^{tb}$ is the translation mobility matrix, $\mathbf{F}$ is the applied force, and $\Theta$ is a vector of three uncorrelated gaussian numbers with zero mean and unit variance. The rotational motion of each SPB is described by the change in its orientation quaternion

$$q_a(t+\Delta t) - q_a(t) = B_{a\alpha}\mu^{rb}_{\gamma\delta}A_{\gamma\beta}T^s_\gamma\Delta t + B_{a\alpha}(\sqrt{\mu^{rb}})_{\alpha\beta}\Theta^q_\beta\sqrt{2k_BT\Delta t} + \lambda_q q_\alpha, \tag{16}$$

where **B** is a matrix described by the elements of the quaternion

$$\mathbf{B} = \frac{1}{2q^4}\begin{pmatrix} q_0 & -q_1 & -q_2 & -q_3 \\ q_1 & q_0 & -q_3 & q_2 \\ q_2 & q_3 & q_0 & -q_1 \\ q_3 & -q_2 & q_1 & q_0 \end{pmatrix}, \tag{17}$$

and $\mathbf{T}^s$ is the torque in the lab coordinate frame on the SPB, $\Theta$ is a vector of three uncorrelated gaussian numbers with zero mean and unit variance, and $\lambda_q$ is a Lagrange multiplier satisfying the condition

$$\lambda_q^2 + 2\lambda_q\mathbf{q}(t)\cdot\widetilde{\mathbf{q}}(t+\Delta t) + \widetilde{\mathbf{q}}^2(t+\Delta t) = 1, \tag{18}$$

where $\widetilde{\mathbf{q}}(t+\Delta t)$ is the quaternion after an unconstrained time step in $\Delta t$ (*Ilie et al., 2015*). We implemented these equations using the Armadillo C++ framework for linear algebra (*Sanderson and Curtin, 2016*; *Sanderson and Curtin, 2019*).

In previous work we modeled the interaction between MTs and a deformable nuclear envelope (*Rincon et al., 2017*; *Lamson et al., 2019*). Here, we use this same force model to describe the interactions between MT plus-ends and the nuclear envelope, and a similar force between SPBs and the nuclear envelope. This force takes on the form in the linear regime of

$$F_{lin}(L) = \frac{F_w}{R_{tube}(ln(2) - \gamma)}L \tag{19}$$

where $L$ is the distance the SPB (or MT) protrudes from the wall, and $F_w$ is the asymptotic wall force, $\gamma$ is Euler's constant, and $R_{tube}$ is the characteristic membrane tube radius. The non-monotonic regime is governed by the equation

$$F_{asymp}(L) = 2aF_w e^{\frac{-L}{b}}cos(\frac{L}{b} + c) + F_w \tag{20}$$

where $a = 0.5416\ldots$ is an integration constant, $b$ is $\sqrt{2R_{tube}}$, and $c = 4.038\ldots$ These two equations can be added together, multiplying the non-monotonic equation by a factor of $(1 - e^{-L})$ to correct the boundary condition at $L = 0$ (*Rincon et al., 2017*; *Lamson et al., 2019*). For SPBs, this force is exerted when they are moved away from the preferred radius of the NE $R$, and only in the radial direction. In addition, we implemented a reorientation torque that causes the SPBs to prefer pointing into the nucleus of the form

$$\mathbf{T}_{\text{SPB,NE,i}} = -\kappa_{\text{r,SPB,NE}}(\hat{\mathbf{u}}_i \cdot \hat{\mathbf{r}}_i + 1)(\hat{\mathbf{u}}_i \times \hat{\mathbf{r}}_i) \tag{21}$$

where $\kappa_{\text{SPB,NE}}$ is the angular spring constant of this interaction. MT minus-ends no longer interact with the nuclear envelope, instead only interacting through their tethers to SPBs.

The soft nuclear envelope model requires the translation and rotation mobility matrices describing the motion of SPBs ($\mu_{\text{SPB,tb}}$ and $\mu_{\text{SPB,rb}}$). These are based on the diffusion of SPBs (*Table 1*). The wall force is described by a membrane tube radius $f_{\text{tube}}$ and asymptotic wall force for both MTs $f_{\text{MT,w}}$ and SPBs $f_{\text{SPB,w}}$ (*Lamson et al., 2019*). SPB-MT tether spring constants were increased to stiffen the interaction between MT minus-ends and the SPBs.

### 1.1.5 Motors and crosslinkers

We model kinesin-5 motors (Cut7), kinesin-14 motors (Pkl1 and Klp2), and crosslinkers (Ase1). Kinesin-5 motors in the model are plus-end directed only when crosslinking antiparallel MTs; otherwise, they are minus-end directed (*Blackwell et al., 2017a*; *Roostalu et al., 2011*; *Gerson-Gurwitz et al., 2011*; *Thiede et al., 2012*; *Fridman et al., 2013*; *Edamatsu, 2014*; *Singh et al., 2018*). Kinesin-14 motors are minus-end directed (*Pidoux et al., 1996*; *Troxell et al., 2001*; *Chen et al., 2012*; *Olmsted et al., 2014*; *Hepperla et al., 2014*;

*Yukawa et al., 2015*; *Yukawa et al., 2018*). Crosslinkers have an increased binding affinity for antiparallel MTs (*Yamashita et al., 2005*; *Kapitein et al., 2008*; *Braun et al., 2011*; *Lansky et al., 2015*). Motors move directionally with a force-dependent velocity based on their stall force, and both motor and crosslinker heads diffuse along MTs while bound (*Table 3*).

The number of active motors and crosslinkers in the model is constrained by experimental data, which estimated total molecule numbers by mass spectrometry and found that mitotic fission-yeast cells have on average 1610 Cut7 tetramers, 2440 Pkl1 and Klp2 dimers (combined), and 3613 Klp9 tetramers and Ase1 dimers (combined) (*Carpy et al., 2014*). We considered these numbers as upper bounds, because of the molecules present in the cell, many may not be active in the spindle because they are outside the nucleus, inactive, and/or in the process of being produced or degraded. We therefore allowed the number of active molecules to vary with the experimental values as an upper bound.

Motors and crosslinkers exert forces and torques on MTs when two heads are bound to two different MTs. The harmonic potential for doubly-bound motors and crosslinkers is

$$u_m(\mathbf{r}_{\mathrm{MT},i}, \mathbf{r}_{\mathrm{MT},j}) = \frac{1}{2} K_{m,0} \left( \left| \mathbf{r}_{\mathrm{MT},j} + \left( s_j - \frac{L_j}{2} \right) \hat{\mathbf{u}}_{\mathrm{MT},j} - \mathbf{r}_{\mathrm{MT},i} - \left( s_i - \frac{L_i}{2} \right) \hat{\mathbf{u}}_{\mathrm{MT},i} \right| - R_{m,0} \right)^2, \quad (22)$$

where $s_i$ and $s_j$ denote the motor/crosslinker head location on MTs $i$ and $j$, $L_i$ and $L_j$ denote the MT lengths, and $R_{m,0}$ is the rest length of the spring. This potential determines the rate of binding/unbinding of crosslinkers in the singly-bound to doubly-bound state. The motors and crosslinkers do not interact with one another.

MT dynamic instability is altered by doubly bound crosslinkers (*Bratman and Chang, 2007*; *Bieling et al., 2010*). We change the dynamic instability parameters when a motor or crosslinker is within the threshold distance $s_l$ of the plus-end of the MT according to

$$
\begin{aligned}
f_c &= f_{c,0} s_{fc}, \\
f_r &= f_{r,0} s_{fr}, \\
v_g &= v_{g,0} s_{vg}, \\
v_s &= v_{s,0} s_{vs},
\end{aligned}
\qquad (23)
$$

where $f_{c,0}$, $f_{r,0}$, $v_{g,0}$ and $v_{s,0}$ are the rates/speeds, and $s_{f/v}$ are the scaling factors. These scaling factors are determined by optimization which matches model to experiment.

Motor and crosslinker proteins bind to/unbind from MTs. Binding from solution is treated as in previous work (*Blackwell et al., 2017a*). Unbound motors and crosslinkers proteins diffuse through the nucleus according to the equation of motion

$$\mathbf{x}(t + \delta t) = \mathbf{x}(t) + \delta \mathbf{x}(t), \qquad (24)$$

where the proteins diffuse in the nuclear volume with diffusion constant $D_{\mathrm{free}}$. Upon reaching the nuclear envelope, motor and crosslinker proteins reflect inward into the nuclear volume.

Once a motor/crosslinker is within a distance of $R_{\mathrm{cap}}$ of the MT, it can bind one head according to the on rate

$$k_{01}(\mathbf{r}_m, \mathbf{r}_{\mathrm{MT}}, \mathbf{u}_{\mathrm{MT}}) = K_a^i \frac{3 \epsilon k_0^{s,i}}{4 \pi R_{\mathrm{cap}}^3} \alpha l_{in}(\mathbf{r}_m, \mathbf{r}_{\mathrm{MT}}, \mathbf{u}_{\mathrm{MT}}), \qquad (25)$$

where $K_a^i$ is the association constant of head $i$, $\epsilon$ is the linear binding site density of an MT, $k_0^{s,i}$ is the turnover rate for protein head $i$ in the singly to unbound transition, $R_{\mathrm{cap}}$ defines the radius of the binding sphere for the transition, $\alpha$ is a scaling factor for the weak dependence of the rate on the total filament length (*Blackwell et al., 2017a*), and $l_{in}$ is the length of the filament defined by $\mathbf{r}_{\mathrm{MT}}$ and $\mathbf{u}_{\mathrm{MT}}$ lying within $R_{\mathrm{cap}}$ of the crosslinker at position $\mathbf{r}_m$. In our simulations $K_a^i$ and $\epsilon$ are multiplied together. Singly bound motor/crosslinker heads detach at a constant rate

$$k_{10} = k_0^{s,i}\alpha, \tag{26}$$

where the $\alpha$ is the same scaling factor used above.

The binding of the second motor/crosslinker head to nearby MTs is force dependent because of the stretch/compression of the tether spring. Detachment from the doubly bound state occurs at rate

$$k_{21}(\mathbf{r}_a, \mathbf{r}_b) = k_{1,\alpha} \exp\left[\beta x_c K_m(|\mathbf{r}_b - \mathbf{r}_a| - R_{m,0})\right], \tag{27}$$

where $k_{21}$ is the off-rate, $k_{1,\alpha}$ is the base rate, $\mathbf{r}_a$ and $\mathbf{r}_b$ are the locations of the motor or crosslinker heads, $\beta$ is the inverse temperature, $x_c$ is the characteristic distance describing force-dependent off-rates, $K_m$ is the motor/crosslinker spring constant, and $R_{m,0}$ is the rest length of the spring. The corresponding on rate is

$$k_{12}(\mathbf{r}_a, \mathbf{r}_{MT}, \mathbf{u}_{MT}) = k_{1,\alpha} c_2 \int \exp\left[-\frac{\beta K_m}{2}(|\mathbf{r}(s)| - R_{m,0})^2 + \beta x_c K_m(|\mathbf{r}(s)| - R_{m,0})\right] ds, \tag{28}$$

where $c_2$ is the effective binding concentration, and $\mathbf{r}(s)$ is the distance between the already bound motor head position $\mathbf{r}_a$ and the position on the second MT denoted by the linear variable $s$,

$$\mathbf{r}(s) = \mathbf{r}_{MT} + s\hat{\mathbf{u}}_{MT} - \mathbf{r}_a, \tag{29}$$

where $\mathbf{r}_{MT}$ is the center of mass of the MT filament, $\hat{\mathbf{u}}_{MT}$ is the orientation of the MT, and $s$ is the linear distance of the second crosslinker head.

## 1.1.6 Chromosomes

Chromosomes contain the genetic material of the cell whose segregation is the primary purpose of mitosis. Sister chromatids are held together by cohesin (*Gay et al., 2012*; *Stephens et al., 2013*; *Pidoux and Allshire, 2004*). Each duplicated sister chromatid assembles the kinetochore onto the centromeric DNA region during mitosis. The outer kinetochore forms the primary MT attachment site for the chromosomes through the KMN (or in yeast, MIND) networks/complexes (*McIntosh et al., 2013*; *Liu et al., 2005*; *Sanchez-Perez et al., 2005*; *Maiato et al., 2004*; *Musacchio and Desai, 2017*; *Cheeseman et al., 2006*; *Foley and Kapoor, 2013*). This network/complex contains the Ndc80, KNL1, Mis12, and Dam/DASH proteins/complexes, and is also important for kinetochore signaling and lost kinetochore recapture (*Dhatchinamoorthy et al., 2017*; *Franco et al., 2007*; *Kalinina et al., 2013*). Chromosomes and kinetochores also contain Aurora B kinase (Ark1 in *S. pombe*), an essential spindle checkpoint component. Aurora B destabilizes incorrect attachments found between the kinetochore and MTs when the chromosome is mis-aligned (*Cheeseman et al., 2002*; *Cimini et al., 2006*; *Koch and Subramanian, 2011*; *Lampson and Cheeseman, 2011*; *Liu et al., 2009*; *Liu et al., 2010b*).

Chromosomes are modeled as sister pairs of chromatids, centromeric DNA, and kinetochores, attached to each other prior to anaphase via a spring potential. We assume that chromosomes do not interact with particles in the spindle, except through the binding/unbinding of attachments at kinetochores, steric repulsion with the nuclear envelope and MTs. A kinetochore moves translationally as a sphere in a viscous medium

$$\mathbf{x}(t+\delta t) = \mathbf{x}(t) + \frac{1}{\gamma_{KC,t}}\mathbf{F}(t)\delta t + \delta\mathbf{x}(t), \tag{30}$$

where $\mathbf{F}(t)$ is the applied force, $\gamma_t$ is the translational drag of the kinetochore, and $\delta\mathbf{x}(t)$ is normally distributed random noise with variance $\langle\delta\mathbf{x}(t)\delta\mathbf{x}(t)\rangle = 2D_{KC}\mathbf{I}\delta t$, and $D_{KC}$ is the diffusion coefficient of a lost kinetochore (*Kalinina et al., 2013*). Kinetochores have principal axes that define their orientation with unit vectors $\hat{\mathbf{u}}$ the outward facing normal of the kinetochore, $\hat{\mathbf{v}}$

along the long arm of the centromeric DNA, and $\hat{\mathbf{w}}$ perpendicular to these (along the short edge of the kinetochore). The equations of motions are

$$\hat{\mathbf{u}}_i(t+\delta t) = \hat{\mathbf{u}}_i(t) + \frac{1}{\gamma_{KC,r}}\mathbf{T}(t) \times \hat{\mathbf{u}}_i(t)\delta t + \delta\hat{\mathbf{u}}_i(t), \tag{31}$$

where $i$ denotes the unit vector in $(\hat{\mathbf{u}}, \hat{\mathbf{v}}, \hat{\mathbf{w}})$, $\mathbf{T}(t)$ is the torque on the kinetochore, and two random Gaussian noise terms are added to $\hat{\mathbf{v}}$ and $\hat{\mathbf{w}}$ with variance

$$\langle\delta\hat{\mathbf{u}}_i(t)\delta\hat{\mathbf{u}}_i(t)\rangle = \frac{2k_BT}{\gamma_{KC,r}}(\hat{\mathbf{I}} - \hat{\mathbf{u}}\hat{\mathbf{u}})\delta t. \tag{32}$$

Kinetochores experience steric repulsion via the WCA potential with the nuclear envelope with a potential

$$u_{\text{wca,KC}}(r_{\min}) = \begin{cases} 4k_BT\left[\left(\frac{\sigma}{r_{\min}}\right)^{12} - \left(\frac{\sigma}{r_{\min}}\right)^6\right] + k_BT, & r_{\min} < 2^{1/6}\sigma, \\ 0, & r_{\min} \geq 2^{1/6}\sigma, \end{cases} \tag{33}$$

where $r_{\min}$ is the minimum distance between the center of the kinetochore and the enclosing sphere of radius $R + (\sigma_{\text{KC}}/2)$. The chromatin does not interact with the nuclear envelope in the model.

Kinetochore plaques are two-dimensional, with long axis $L_{KC,0}$ along the centromeric DNA region and short axis $L_{KC,1}$ perpendicular to this region. Because MTs were not observed to pass through kinetochores in fission yeast spindle tomographic reconstructions (***Ding et al., 1993***), we included a steric repulsion between the plaques and MTs of the form

$$u_{\text{wca,MT-KCmesh}}(r_{\min}) = \begin{cases} 4k_BT\left[\left(\frac{\sigma}{r_{\min}}\right)^{12} - \left(\frac{\sigma}{r_{\min}}\right)^6\right] + k_BT, & r_{\min} < 2^{1/6}\sigma, \\ 0, & r_{\min} \geq 2^{1/6}\sigma, \end{cases} \tag{34}$$

where $r_{\min}$ is the minimum distance from the MT to the triangulated kinetochore mesh, and $\sigma$ defines half of the MT diameter to approximate an infinitely thin kinetochore. This force contributes to force-induced catastrophe when the MT tip interacts with the kinetochore.

The centromeric DNA regions is modeled as a spherocylinder with length $r_{\text{centromere}}$ and diameter $d_{\text{centromere}}$. Kinetochore plaques are located on the surface of these regions, with an offset from the center of the centromeric DNA chromatid of $r_{\text{KC-cen}}$. Centromeric DNA regions experience a weak repulsive interaction with MTs of the form

$$u_{\text{gauss}}(r_{\min}) = \frac{A_{CMT}}{\sigma\sqrt{2\pi}}\exp\left[\frac{-r_{\min}^2}{2\sigma^2}\right], \tag{35}$$

where $\sigma = d_{\text{centromere}}/10 + \sigma_{\text{MT}}/10$, $A_{CMT}$ sets the maximum repulsion, and $r_{\min}$ is the minimum distance between the chromatin spherocylinder and the MT spherocylinder. The strength of this potential is set on the order of 1 $k_BT$, and contributes to MT force-induced catastrophe.

Sister chromosomes, chromatids, and kinetochores are bound to each other until anaphase by linear and angular springs. Each centromeric DNA region has a right-handed coordinate system that is determined at the beginning of the simulation, and defines the principle axes of the chromatid/centromeric DNA region/kinetochore $(\hat{\mathbf{u}}_i, \hat{\mathbf{v}}_i, \hat{\mathbf{w}}_i)$, where $i$ now labels the sister of the pair. For the interkinetochore spring, $\hat{\mathbf{u}}_i$ is the outward-facing normal of the first kinetochore, and the inward-facing normal of the second kinetochore, and $\hat{\mathbf{v}}_i$ points along the chromatid arm. The potential is

$$u_{\text{chromosome}} = \frac{1}{2}\kappa_C(r - R_{C,0})^2 + \frac{1}{2}\kappa_{C,u}(\theta_A^2 + \theta_B^2) + \frac{1}{2}\kappa_{C,v}\theta_v^2, \tag{36}$$

where $\mathbf{r} = \mathbf{r_A} - \mathbf{r_B}$, $r = |\mathbf{r}|$, $cos(\theta_{A,B}) = \hat{\mathbf{u}}_{A,B}\cdot\hat{\mathbf{r}}$ and $cos(\theta_v) = \hat{\mathbf{v}}_A\cdot\hat{\mathbf{v}}_B$. This potential serves to align the sister kinetochores/chromatids so that they are back-to-back with inter-kinetochore distance $R_{C,0}$ and aligning spring constants $\kappa_C$, $\kappa_{C,u}$, and $\kappa_{C,v}$.

The forces and torques on the chromatids due to the interkinetochore potential (*Equation 36*) is computed as in previous work (*Allen and Germano, 2006*). The force on chromatid A is

$$\mathbf{f}_A = -\kappa_C(r - R_{C,0})\hat{\mathbf{r}} - \frac{\kappa_{C,u}}{r}\left[\frac{\theta_A}{\sin(\theta_A)}(\hat{\mathbf{r}} \times (\hat{\mathbf{r}} \times \hat{\mathbf{u}}_A)) + \frac{\theta_B}{\sin(\theta_B)}(\hat{\mathbf{r}} \times (\hat{\mathbf{r}} \times \hat{\mathbf{u}}_B))\right]. \tag{37}$$

The force on chromatid B is equal and opposite. The torques are

$$\boldsymbol{\tau}_A = -\kappa_{C,u}\left[\frac{\theta_A}{\sin(\theta_A)}(\hat{\mathbf{r}} \times \hat{\mathbf{u}}_A)\right] + \kappa_{C,v}\left[\frac{\theta_v}{\sin(\theta_v)}(\hat{\mathbf{v}}_A \times \hat{\mathbf{v}}_B)\right], \tag{38}$$

$$\boldsymbol{\tau}_B = -\kappa_{C,u}\left[\frac{\theta_B}{\sin(\theta_B)}(\hat{\mathbf{r}} \times \hat{\mathbf{u}}_B)\right] - \kappa_{C,v}\left[\frac{\theta_v}{\sin(\theta_v)}(\hat{\mathbf{v}}_A \times \hat{\mathbf{v}}_B)\right]. \tag{39}$$

These can be checked for validity by using $\mathbf{r} \times \mathbf{f_A} + \boldsymbol{\tau_A} + \boldsymbol{\tau_B} = \mathbf{0}$.

There are 3 ($N_{AF}$) kinetochore-MT binding sites on average in fission yeast with inter-binding site spacing $r_{AF,ex}$ of 40 nm (*Ding et al., 1993*). Kinetochore-MT attachments are modeled as a linear and angular spring

$$u_{AF} = \frac{1}{2}\kappa_m(r(s) - r_0)^2 + \frac{1}{2}\kappa_r(\hat{\mathbf{u}}_{KC} \cdot \hat{\mathbf{f}}(s) - 1)^2, \tag{40}$$

where $\kappa_m$ is the linear spring constant, $r_0$ is the length of the attachment factor, $\kappa_r$ is the angular spring constant, and $\hat{\mathbf{u}}_{KC}$ is now the outward-facing normal orientation of the kinetochore. The vector $\mathbf{r}(s)$ is the distance from the kinetochore binding site location on the kinetochore to the attachment site on the MT

$$\mathbf{r}(s) = \mathbf{r}_{MT} + s\hat{\mathbf{u}}_{MT} - \mathbf{r}_{AF}, \tag{41}$$

where $\mathbf{r}_{AF}$ is the coordinate of the kinetochore binding site. Forces and torques from this potential are also calculated according to *Allen and Germano (2006)*, with the force on the bound MT from the kinetochore

$$\mathbf{f}_{MT} = -\kappa_m(r - r_0)\hat{\mathbf{f}} + \frac{\kappa_r}{r}(\hat{\mathbf{u}}_{KC} \cdot \hat{\mathbf{f}} - 1)[\hat{\mathbf{f}} \times (\hat{\mathbf{f}} \times \hat{\mathbf{u}}_{KC})], \tag{42}$$

where $r = |\mathbf{r}(s)|$. The torque on the kinetochore is

$$\boldsymbol{\tau}_{KC} = \kappa_r(\hat{\mathbf{u}}_{KC} \cdot \hat{\mathbf{f}} - 1)(\hat{\mathbf{f}} \times \hat{\mathbf{u}}_{KC}). \tag{43}$$

Kinetochore-MT attachments have been shown to have force-dependent lifetime (*Akiyoshi et al., 2010*; *Miller et al., 2016*). The on rate for kinetochores binding MTs is analogous to that of motor and crosslinker proteins with an additional the angular term

$$k_{on} = k_0 c_{on} \int \exp\left[-\frac{\beta\kappa_m}{2}(r(s) - r_0)^2 + x_c\beta\kappa_m(r(s) - r_0) - \frac{\beta\kappa_r}{2}(\hat{\mathbf{u}}_{KC} \cdot \hat{\mathbf{f}} - 1)^2 - \chi_c\beta\kappa_r(\hat{\mathbf{u}}_{KC} \cdot \hat{\mathbf{f}} - 1)\right]ds, \tag{44}$$

where $x_c$ is the linear characteristic distance of the force-dependent interaction, and $\chi_c$ controls the angular force dependence. Notice that for the $\chi_c$ enhancement to the angular rate, we are choosing the more numerically stable factor of $f_\theta = -\chi_c K_r(\hat{\mathbf{u}}_{KC} \cdot \hat{\mathbf{f}} - 1)$.

The accompanying off rate is

$$k_{\text{off}} = k_0 \exp[x_c\beta\kappa_m(r - r_0)] \exp[-\chi_c\beta\kappa_r(\hat{\mathbf{f}} \cdot \hat{\mathbf{u}}_{KC} - 1)], \tag{45}$$

where $r = |\mathbf{r}|$ is the distance between the binding site location on the kinetochore and the MT binding location, and $\hat{\mathbf{r}}$ is the orientation of this separation. These rates are only calculated every $N_{kmc}$ steps because of the complexity involved in their evaluation.

Kinetochores affect MT dynamic instability in a force-dependent manner when attached to MT plus-ends. This affects the growing speed, shrinking speed, rescue frequency and catastrophe frequency in the form

$$k(F) = k_0 \exp\left[\frac{F}{F_c}\right], \tag{46}$$

where $F_c$ is the characteristic force, and $k(F)$ and $k_0$ are the force-dependent and base speed/ frequency.

Kinetochores can bind both MT lateral walls and plus-ends with different binding affinity ($c_{AF,tip}$ the effective concentration for the plus-end, $c_{AF,side}$ the effective concentration for MT lateral wall, $k_{AF,tip,a}$ the rate for attaching to an assembling MT tip, $k_{AF,tip,d}$ the rate for attaching to a disassembling MT tip, and $k_{AF,side}$ the rate for attaching to the MT wall, *Table 5*; *Asbury et al., 2006*; *Akiyoshi et al., 2010*; *Kalinina et al., 2013*). The tip region of the MT is defined by $l_{AF,tip}$, and only kinetochores bound in this region can affect MT dynamic instability. Attachments bound to the tip have MT-polymerization-state-dependent lifetime. We require that only one attachment factor can bind to the same MT tip ($b_{AF,tip}$), and so if two or more are found bound to the same tip, the attachment factor farther from the tip is unbound.

Progressive restriction of kinetochore-MT attachment is modeled by changing the angular spring constant based on the number of bound MTs

$$\kappa_r = \kappa_r(N_a), \tag{47}$$

where $\kappa_r$ is the angular spring constant and $N_a$ the number of bound MTs to the kinetochore. Note that each kinetochore can have a different number of attachments, and there is an angular spring constant for unbound kinetochores that controls the binding rate of lost kinetochores.

Kinetochore attachments are mobile on MTs, and they diffuse along MTs, track MT ends when attached at the tip, and can have directed motion (*Wood et al., 1997*; *Akera et al., 2015*). These attachments move on MTs according to

$$x_l(t+\delta t) = x_l(t) + v(\mathbf{F}, \mathbf{u}_{MT})\delta t + \beta D_l \mathbf{F} \cdot \mathbf{u}_{MT}\delta t + f_{AF,track}d_{poly} + \delta x_l(\delta t), \tag{48}$$

where $l_{AF,tip}$ denotes if the attachment is in the MT tip region, $\mathbf{F}$ is the force on the attachment, $\mathbf{u}$ is the orientation of the MT, $D_l$ is the one-dimensional diffusion coefficient of the attachment on the MT ($D_{tip}$ for MT tips, $D_{side}$ for sides), $f_{AF,track}$ is the tip-tracking fraction, and $d_{poly}$ is the distance the MT polymerized in the last time step (this effect is only turned on for MT tips). The random displacement term $\delta x_l(t)$ is Gaussian random noise with variance $2D_l\delta t$. Attachments between kinetochores and MTs do not detach when they reach MT tips. The kinetochore motor force-velocity relation is linear, as for crosslinking motors,

$$v(\mathbf{F}, \mathbf{u}_{MT}) = v_0 \max\left(0, \min\left(1, 1 + \mathbf{u}_{MT} \cdot \frac{\mathbf{F}}{f_{stall}}\right)\right), \tag{49}$$

where $f_{stall}$ is the stall force of the attachment and $v_0$ is the speed.

When an attachment factor is bound near an MT tip, the MT dynamics are destabilized by a combination of proteins, and this is represented in the model by the enhanced catastrophe factor $s_{fc,dam1}$. This has the effect of multiplying the base catastrophe rate by this constant.

## 1.1.7 Kinetochore-MT attachment error correction models: biorientation-dependent and force-dependent

In the initial formulation of the model, we implemented a rule that kinetochore-MT attachments in the model are destabilized when the chromosome is not bioriented. In this case, each attachment and detachment rate is multiplied by the factor $s_{k,ABK}$ to maintain the binding equilibrium between the on- and off-rates

$$k_{AF} = \begin{cases} k_{AF}, & \text{amphitelic,} \\ s_{ABK}k_{AF}, & \text{misaligned,} \end{cases} \tag{50}$$

where $k_{AF}$ is the rate of the kinetochore-MT attachment or detachment and $s_{ABK}$ is the misaligned destabilization factor.

To make the error-correction model more mechanistic, we also tested a version of force-dependent error correction, building on previous results that show that kinetochore-MT attachments are stabilized by force (**Nicklas and Koch, 1969**; **Cane et al., 2013**). We made the kinetics of kinetochore-MT attachments dependent on interkinetochore tension in the form

$$k_{AF} = \frac{k_{AF,0}}{1 + \frac{F}{F_{EC,0}}} \quad when \, \mathbf{F} \cdot \mathbf{u}_{KC} < 0, \tag{51}$$

where $F$ is the interkinetochore force and $F_{EC,0}$ is a characteristic force for significant stabilization: when the interkinetochore tension reaches the value $F_{EC,0}$, the rate drops by a factor of two from its unstabilized value. Therefore, smaller values of $F_{EC,0}$ correspond to higher force sensitivity. This stabilization is only active when the force on the kinetochore is in the opposite direction of the kinetochore outward facing normal orientation. Only kinetochore-MT attachment off rates are reduced when there is interkinetochore tension.

We carried out several rounds of optimization for the force-dependent error correction model, as initial models did not lead to biorientation. Recent work has shown that kinetochores may experience tension before biorientation (**Kuhn and Dumont, 2017**), and so we varied additional parameters in the model to facilitate tension generation prior to biorientation. These parameters were the angular spring constants of the interkinetochore spring ($\kappa_{C,u}$ and $\kappa_{C,v}$), the characteristic angular factor for binding high angles to the kinetochore plate $\chi_C$, the effective concentration for binding to lateral walls of MTs $c_{AF,side}$, and the number of kinesin-5 motors $N_{K5}$, which affect overall spindle force generation. We identified model parameters that favored end-on over lateral attachments, inhibited attachments at high angle, and allowed sister kinetochores to more easily reorient (**Table 6**).

## A.1.8 Anaphase

Anaphase is triggered by waiting until all three chromosomes are bioriented simultaneously for a time $\tau_{SAC}$. Then the potential connecting the two sister chromatids is removed, and misaligned destabilization is turned off ($s_{k,ABK} = 1$). The rates governing kinetochore-MT attachment are modified to all be the same value ($k_{AF,anaphase}$), and kinetochore MTs are forced to undergo depolymerization. Finally, the MT shrinking speed is changed to be $v_{anaphase,s,0}$.

## A.1.9 Initial conditions

At the beginning of mitosis in *S. pombe* the two SPBs are linked by a bridge (**Bouhlel et al., 2015**; **Lee et al., 2014**). Initially the spindle pole bodies are placed adjacent with their center separated by the spindle pole body diameter plus the bridge size $\sigma_{SPB} + 75$ nm. MTs are inserted randomly onto each SPB so that they do not overlap and are within the nuclear volume. Initially MTs are their minimum length (75 nm) and in the paused dynamic instability state. Motors and crosslinkers are randomly inserted into the nucleus. Chromosomes are placed near the spindle pole bodies, with a single attachment between one kinetochore and the first spindle pole body. Simulations are started with SPBs fixed for a linkage time $\tau_{link}$.

## Parameter constraint and model refinement

The constrain unmeasured parameters, we performed refinement and optimization, based on previous work (**Blackwell et al., 2017a**). To do this, we measured spindle length and

movement of a single sister kinetochore pair for 9 cells (as discussed in experimental methods below). The fitness function defined to compare simulation and experiment includes three contributions. (1) Spindle structure fitness is based on spindles reconstructed from electron tomography. (2) The dynamics of spindle length, kinetochore movement, and interkinetochore separation were compared to fluorescence microscopy. (3) We sought to maximize the amount of time all chromosomes are bioriented.

The spindle structural parameters were similar to those used in previous work (**Blackwell et al., 2017a**). The length distribution of MTs, the length distribution of interpolar MTs, the maximum pairing length, and the angular distribution of MTs relative to the spindle axis from three different tomographic reconstructions of fission-yeast spindles of different lengths are compared to simulation results. Spindles matching the target length ±50 nm are used to determine distributions from simulation. All measurements from all runs at a particular parameter point are compiled into one distribution for comparison with tomographic data. The EM fitness is defined as

$$
f_{\mathrm{EM},i,l} = \begin{cases} -10, & p = 0, \\ \frac{log_{10}(p_{i,l})}{100}, & p > 0, \end{cases}
\tag{52}
$$

where $i$ labels the distribution and $l$ the target length. The p-value from the Kolmogorov-Smirnov (KS) test of the combined measurements in the model is used as the input to this function. The total EM fitness is the average

$$
f_{\mathrm{EM}} = \frac{1}{3} \sum_{\mathrm{length}} \frac{1}{4} \sum_{\mathrm{distribution}} f_{EM,i,l}.
\tag{53}
$$

The value of this objective function lies in $(-10, 0)$, where a larger value indicates a better match.

We measured spindle length, kinetochore position, and interkinetochore distance. To quantify similarity between simulated and experimental measurements, we computed the Pearson correlation coefficient with simulation data sampled at same time as experimental measurements. Spindle length fitness is

$$
f_{\mathrm{L}}(s,t) = \rho_{\mathrm{L}}(s,t),
\tag{54}
$$

where $s$ labels the simulation, $t$ labels a distinct experimental trace of spindle length versus time, and $\rho$ is the Pearson correlation coefficient. When comparing the dynamics of kinetochore distance from a single spindle pole, we find the maximum Pearson correlation coefficient to determine which spindle pole to use in the analysis. The spindle pole body-kinetochore fitness is

$$
f_{\mathrm{SPB-KC}}(s,t) = \frac{1}{2N_c} \sum_c \max(\rho_{1,1,c} + \rho_{2,2,c}, \rho_{1,2,c} + \rho_{2,1,c}),
\tag{55}
$$

where $N_c$ is the number of chromosomes, $c$ sums over the chromosomes, and $\rho_{1,1,c}$ is the Pearson correlation coefficient for comparing kinetochore one in the simulation to kinetochore one in the experiment, and so on. The interkinetochore separation has fitness

$$
f_{\mathrm{KC-KC}}(s,t) = \frac{1}{N_c} \sum_c \rho_{\mathrm{KC-KC},c},
\tag{56}
$$

where $\rho_{\mathrm{KC-KC},c}$ is the Pearson correlation coefficient of interkinetochore separation of chromosome $c$. The dynamic fitness function is then

$$
f_{\mathrm{d}}(s,t) = f_{\mathrm{L}} + f_{\mathrm{SPB-KC}} + f_{\mathrm{KC-KC}}.
\tag{57}
$$

For each set of simulation parameters, the dynamic fitness is averaged over all simulated and experimental time traces.

To promote long-lived simultaneous biorientation of all chromosomes and end-on kinetochore attachments to MTs, we measure the fraction of simultaneous biorientation

$$f_{\mathrm{I}} = \frac{\sum_i f_a^1(i) f_a^2(i) f_a^3(i) [L(i) > 1\,\mu m]}{\sum_i 1}, \tag{58}$$

where $i$ is the time, $f_a^c(i)$ is one if chromosome $c$ has amphitelic attachment at time $i$, and $L(i)$ is the spindle length at time $i$. This value is larger when all three chromosomes are simultaneously bioriented for longer time. Next we measure the weighted average number of attachments

$$f_{\mathrm{b}} = \frac{\sum_{i,c} f_a^c(i) N_a^c(i)}{\sum_{i,c} N_{\max}}, \tag{59}$$

where $N_a^c$ is the number of end-on attachments of chromosome $c$ at time $i$ and $N_{\max}$ is the maximum number of kinetochore attachments per chromosome at time $i$ (six per chromosome).

The total fitness is the weighted sum

$$f = f_d + f_{\mathrm{EM}} + 2f_{\mathrm{I}} + 2f_{\mathrm{b}}. \tag{60}$$

Here $f_d$ can take values $(-3, 3)$, $f_{\mathrm{EM}}$ $(-10, 0)$, and $f_{\mathrm{I}}$ and $f_{\mathrm{b}}$ $(0, 1)$, which are weighted in the total fitness to $(0, 2)$. The total fitness therefore falls in the range $(-13, 7)$. The reference model has a total fitness of 3.36 with dynamic fitness 1.23, EM fitness $-0.10$, fraction simultaneous biorientation 0.68, and weighted average number of attachments 0.43. An example of model/experiment comparison is shown in *Appendix 1—figure 2*.

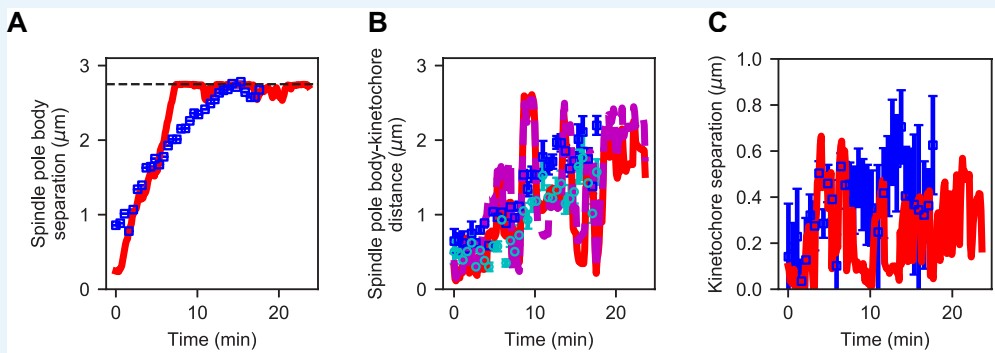

**Appendix 1—figure 2.** Reference model generates similar dynamics of spindle length and kinetochore position compared to experiment. (**A**) Spindle length versus time for experiment (blue) and refined model (red). (**B**) Spindle pole body-kinetochore distance versus time for a single kinetochore pair (Cen2) in experiment (blue, cyan) and refined model (red, magenta). (**C**) Kinetochore separation versus time for experiment (blue) and refined model (red). This comparison gives Pearson correlation coefficients for length = 0.891, SPB-KC distance = 0.72, Interkinetochore distance = 0.42.

### A.2.1 Optimization of parameters

We optimized unknown or poorly constrained parameters, as in previous work (*Blackwell et al., 2017a*). We attempted to use particle-swarm optimization (*Kennedy and Eberhart, 1995*) by first randomly sampling parameter sets, and then refining the parameters to reach maximum fitness. However, for our high-dimensional optimization we found slow convergence, and used human input to guide the particle swarm. This included scans of single parameters identify parameter ranges that increased the total fitness.

Unknown or poorly constrained parameters that we optimized include the stabilization parameters of MTs in bundles and the number and force-sensitivity of the motors and crosslinkers (*Tables 1* and *3*). We note that the characteristic distances found for force-dependent unbinding are similar to previously measured kinesin force-dependence

(*Arpağ et al., 2014*). For the chromosome and kinetochore model, we optimized multiple parameters. The linear and angular spring constants of interkinetochore interactions were initially taken from previous models, then optimized (*Table 4*; *Stephens et al., 2013*; *Gay et al., 2012*). We also optimized the strength of the soft repulsion between chromatin and MTs; the angular spring constants for progressive locking; concentration, rate, and characteristic distance kinetochore-MT attachments (*Akiyoshi et al., 2010*); the movement of kinetochore-MT attachments on MTs; the amount of enhanced catastrophe from attachments at MT plus-ends; and the amount misaligned attachments are destabilized (*Table 5*).

## Experimental methods

The fission-yeast strain includes *cen2-GFP* to label centromeric DNA with lacI-GFP of chromosome 2 (*Appendix 1—table 1*; *Yamamoto and Hiraoka, 2003*). The microtubules were tagged with low-level labeling of *mCherry-atb2* (*Yamagishi et al., 2012*). 9 cells which began in interphase were continuously imaged through anaphase B. The time-lapse images shown in *Figure 2E* and *Figure 6E* were taken using live cell preparation and spinning-disk confocal imaging on a Nikon Eclipse Ti microscope as previously described (*Blackwell et al., 2017a*; *Gergely et al., 2016*). Cell temperature was maintained at 25C with a CherryTemp temperature control device (Cherry Biotech, Rennes, France) with an accuracy of +/- 0.1C. 3D images were obtained with an EM Gain of 300 and an exposure time of 100 ms for the 488 nm laser and 150 ms for the 561 nm laser, both at 100% laser power. 7 planes were acquired in the z dimension with 500 nm separation between each plane. Images are displayed as smoothed maximum-intensity projections with ~8 s between successive images and were prepared using Image J software (NIH, Bethesda, Maryland).

**Appendix 1—table 1.** Strain used in this study.

| Name | Genotype | Notes |
|---|---|---|
| MB 998 | cen2::kanr-ura4[+]-lacOp, his7[+]::lacI-GFP, z:adh15:mCherry-atb2:natMX6, leu1-32, h- | This study |

Analysis of experimental images was performed in MATLAB by extending previous work (*Jaqaman et al., 2008*). Individual cells were segmented using morphology and geometric considerations on time-averaged and space-convolved videos to find locations of objects persisting in both space and time. Using the microtubule channel, only cells that at some point exhibited a bright spindle were segmented. After segmentation, the position of each object was estimated. The first SPB location was estimated to be at the location of the brightest pixel in the image in the MT channel. We estimated spindle orientation by thresholding the image to find the brightest ~10 pixels, and then estimated the spindle axis by the direction of the major axis of the ellipse that encloses the active pixels. The second SPB is assumed to have 80% of the intensity of the first SPB and to lie along the spindle axis. We then estimated a 3D Gaussian line connecting the two SPBs to represent the central MT bundle. Kinetochore positions were estimated by finding peaks in the intensity image in the kinetochore channel. Peaks whose width was comparable to the point spread function were treated as possible kinetochores, and each kinetochore is modeled as a 3D Gaussian.

We fit the position of the objects in the system using a non-linear least squares optimization to minimize the residual error between the raw image and a simulated image using lsqnonlin in MATLAB. This fit varied 13 parameters in the microtubule channel and 13 in the kinetochore channel. Features from multiple time points were tracked. Spindle length was directly determined in each frame, and the two kinetochores were tracked with u-track (*Jaqaman et al., 2008*).

## Simulation snapshots and simulated fluorescence images

We generated simulation snapshots amd simulated fluorescence images by first using a quaternion formulation that aligns view orientation vectors with spindle vectors to obtain planar images of the spindle. The algorithm aligns

$$\hat{\mathbf{r}}_{\text{spindle}} \rightarrow \hat{\mathbf{x}}, \tag{61}$$

$$\hat{\mathbf{n}}_{\text{SPBs}} = \frac{\mathbf{r}_{\text{SPB1}} \times \mathbf{r}_{\text{SPB2}}}{|\mathbf{r}_{\text{SPB1}} \times \mathbf{r}_{\text{SPB2}}|} \rightarrow \hat{\mathbf{z}}, \tag{62}$$

where the spindle axis $\hat{\mathbf{r}}_{\text{spindle}}$ is aligned with the unit orientation vector $\hat{\mathbf{x}}$, and the normal of the two SPB vectors $\hat{\mathbf{n}}_{\text{SPBs}}$ is aligned with $\hat{\mathbf{z}}$ (toward the viewer). Simulated fluorescence images are rotated so that the spindle axis lies along the $\hat{\mathbf{y}}$ vector.

Simulated fluorescence images are created by applying a Gaussian blur to every point of the object of interest. For point-like objects such as kinetochores, we applied a 2D Gaussian with the xy point-spread-function and pixel size measured on the microscope. MT fluorescence uses the convolution of a point-Gaussian with the 2-dimensional line

$$
\begin{aligned}
I(x,y,A,L,\sigma,x_0,y_0,\theta) = \ & A \exp[(y-y_0)\cos(\theta)+(x_0-x)\sin(-\theta)] \\
& \{\text{Erf}[\frac{L+(x_0-x)\cos(\theta)+(y_0-y)\sin(-\theta)}{\sqrt{2}\sigma}]+ \\
& \text{Erf}[\frac{(x_0-x)\cos(\theta)+(y_0-y)\sin(-\theta)}{\sqrt{2}\sigma}],\}
\end{aligned}
$$

where $A$ is the amplitude, $L$ is the length of the line segment, $\sigma$ is the point-spread, $x_0$ and $y_0$ are the starting point of the line segment, and $\theta$ is the 2-dimensional direction of the line segment in the xy-plane.

