## [Decision Letter]

**Acceptance summary:**

Edelmaier et al. describe a computational simulation of mitosis in the fission yeast *Schizosaccharomyces pombe* that includes bipolar spindle assembly, chromosome capture by spindle microtubules, chromosome bi-orientation, and kinetochore-microtubule attachment error correction. Their model provides a comprehensive view of mitotic cell division and proposes a set of rules that helps explain how mitosis is executed in fission yeast and possibly in metazoan cells, whose spindles are likely to be governed by similar fundamental principles.

**Decision letter after peer review:**

Thank you for submitting your article "Mechanisms of chromosome biorientation and bipolar spindle assembly analyzed by computational modeling" for consideration by *eLife*. Your article has been reviewed by Anna Akhmanova as the Senior Editor, a Reviewing Editor, and two reviewers. The reviewers have opted to remain anonymous.

The reviewers have discussed the reviews with one another and the Reviewing Editor has drafted this decision to help you prepare a revised submission.

Summary:

The manuscript by Edelmaier et al. describes a computational simulation of spindle assembly, chromosome alignment, and chromosome segregation in *S. pombe*. The authors combine and extend their previous modeling efforts regarding fission yeast spindle assembly and chromosome bi-orientation to produce a new model that explains well: (1) the initial establishment of spindle bipolarity, (2) spindle stability and elongation during metaphase, and (3) kinetochore capture and attachment by MTs. The model, which encompasses all phases of mitosis in fission yeast, is currently state-of-the-art for the field and represents an important advance. However, several issues were raised by the reviewers which require attention. These are listed below.

Essential revisions:

1) The representation of error correction in the model should be re-examined. The destabilization of incorrect attachments is suggested to represent the activity of Aurora B kinase. However, this would require the enzyme to somehow respond to a highly complex property of the attachment – i.e., is it attached to a microtubule emanating from the correct pole? In comparison to all the other rules included in the simulations, which seem physically plausible, this particular rule seems rather ad hoc, unrealistic. How can an enzyme distinguish correct from incorrect? Simpler ideas have been suggested in the literature that might explain how Aurora B destabilizes incorrect attachments. For example, it might act selectively on kinetochore-microtubule attachments that lack force. The authors should consider this, or other, more mechanistic rules for error correction.

2) The simulated spindles fail to assemble and elongate in the absence of crosslinker Ase1 (Figure 2K). Because a force-balanced constant metaphase spindle length is evident in the cell, and because spindle assembly is achieved without the crosslinker Ase1, the impact as predictive power of the model/simulation is limited. The authors should explore additional simulation parameters that produce Ase1 spindle results in their model that are consistent with published experimental results.

3) The final metaphase spindle lengths observed in the simulations may not be a natural consequence of the force-balance inherent in the spindle, but instead due to the simplifying assumption that the nuclear envelope is defined as a rigid sphere. While the reviewers acknowledge that including flexibility of the nuclear membrane would likely increase the computational requirements of the model significantly, textual revisions acknowledging the limitations of the current model in regard to this point are needed.

4) The model used ~200 Kinesin-5 and Kinesin-14 motors, and ~600 Ase1 crosslinkers for the simulations. These numbers are 5x to 15x less than reported values existing in fission yeast (PomBase). The usage of simulation parameters different from experiment measurements undermines the impact of the model. The authors should either address this in their simulations or explicitly acknowledge the caveat of including lower levels of motors and crosslinking MAPs in the model compared to what has been measured experimentally.

[Editors' note: further revisions were suggested prior to acceptance, as described below.]

Thank you for resubmitting your work entitled "Mechanisms of chromosome biorientation and bipolar spindle assembly analyzed by computational modeling" for further consideration by *eLife*. Your revised article has been evaluated by Anna Akhmanova (Senior Editor) and a Reviewing Editor.

The manuscript has been improved but there are some remaining issues that need to be addressed before acceptance.

In subsection “Single model perturbations recapitulate the requirement for kinesin-5 motors and CLASP”, the authors added extensive revision on the role of Ase1 crosslinker (and pliable nuclear envelop). Specifically, they stated that in their model, the motor Klp9 can also act as a crosslinker in the absence of Ase1. Therefore, the removal of Ase1 would still produce a bipolar spindle, due to the action of Klp9. Reported data from several labs (McCollum, Millars, Toda, Tran) clearly showed that Klp9 does not function until anaphase. Thus, the authors cannot model the absence of Ase1 as still having crosslinking via Klp9.

---

## [Author Response]

Essential revisions:1) The representation of error correction in the model should be re-examined. The destabilization of incorrect attachments is suggested to represent the activity of Aurora B kinase. However, this would require the enzyme to somehow respond to a highly complex property of the attachment – i.e., is it attached to a microtubule emanating from the correct pole? In comparison to all the other rules included in the simulations, which seem physically plausible, this particular rule seems rather ad hoc, unrealistic. How can an enzyme distinguish correct from incorrect? Simpler ideas have been suggested in the literature that might explain how Aurora B destabilizes incorrect attachments. For example, it might act selectively on kinetochore-microtubule attachments that lack force. The authors should consider this, or other, more mechanistic rules for error correction.

We agree with the referee that the implementation of error correction in the original manuscript was ad hoc and assumed nonlocal information, that is, that Aurora B could distinguish incorrect attachments. In response to the referee’s suggestion, we have implemented a force-based error-correction mechanism in which microtubule-kinetochore attachments depend on the interkinetochore force: low/no-force attachments are destabilized, and the attachments become more stable with increasing force. We found that implementing this change required some additional changes to the model for biorientation to occur and be maintained. With these changes, error correction does occur and leads to persistent biorientation in the model, although not for quite as large a percentage of the simulation as occurs for the ad-hoc error correction model. We show as well that error correction works best for a particular value of the force constant that controls how rapidly stabilization turns on as force is increased.

In future work, we would like to examine aspects of error correction in more detail. We believe that the change to include a version of force-dependent error correction greatly improves the manuscript, and thank the referee for the suggestion.

2) The simulated spindles fail to assemble and elongate in the absence of crosslinker Ase1 (Figure 2K). Because a force-balanced constant metaphase spindle length is evident in the cell, and because spindle assembly is achieved without the crosslinker Ase1, the impact as predictive power of the model/simulation is limited. The authors should explore additional simulation parameters that produce Ase1 spindle results in their model that are consistent with published experimental results.

We agree with the referee that this discrepancy with experimental results is concerning. However, we believe that it requires interpretation in light of recent results that the tetrameric motor Klp9 has a non-motor crosslinking function that is important for spindle integrity (Yukawa et al., 2019). In our model, the molecules represent classes of functions like crosslinking, so our crosslinking molecule would include the function of both Ase1 and Klp9 in crosslinking spindle microtubules. Consistent with this idea, Yukawa et al. demonstrated that the double deletion of Klp9 and Ase1 is lethal, as our model shows. Using data from Carpy et al., 2014, we find that there are approximately equal numbers of Klp9 tetramers and Ase1 dimers in fission-yeast cells. Therefore, we estimate that deleting Ase1 corresponds to removing about half the crosslinking molecules in our simulations. We studied how spindle assembly and crosslinking changes with crosslinker number and added this to Figure 3C. We find that in our model losing about half the crosslinkers leads to slightly shorter spindles and slightly less biorientation, consistent with experimental results and cell viability. We have added text to the manuscript to discuss this in more detail.

3) The final metaphase spindle lengths observed in the simulations may not be a natural consequence of the force-balance inherent in the spindle, but instead due to the simplifying assumption that the nuclear envelope is defined as a rigid sphere. While the reviewers acknowledge that including flexibility of the nuclear membrane would likely increase the computational requirements of the model significantly, textual revisions acknowledging the limitations of the current model in regard to this point are needed.

We agree with the referees that the rigidity of the nuclear envelope in our model may affect force balance and spindle length. Therefore, we have extended our model to allow the SPBs to move in three dimensions with a radial potential that mimics the deformation of the nuclear envelope that would be required to move the SPB radially. We have added Figure 3D showing the effects of varying the strength of this nuclear envelope force. For realistic values of the nuclear envelope force of around 17 pN, this soft nuclear envelope leads to slightly longer spindles that are still around the biologically measured value of 3 µm. Therefore, we believe that the nuclear envelope does contribute to spindle force balance, but its contribution is modest compared to motors, crosslinkers, and chromosomes.

4) The model used ~200 Kinesin-5 and Kinesin-14 motors, and ~600 Ase1 crosslinkers for the simulations. These numbers are 5x to 15x less than reported values existing in fission yeast (PomBase). The usage of simulation parameters different from experiment measurements undermines the impact of the model. The authors should either address this in their simulations or explicitly acknowledge the caveat of including lower levels of motors and crosslinking MAPs in the model compared to what has been measured experimentally.

Looking at data from Carpy et al. available in Pombase, we find the number of oligomeric molecules (tetramers of Cut7 and Klp9, dimers of Ase1, Pkl1, and Klp2) are: 1610 Cut7, 2440 Pkl1 and Klp2 combined, and 3613 Klp9 and Ase1 combined. Our simulations use 170-250 kinesin-5 motors, 230 kinesin-14 motors, and 657 crosslinkers. These are indeed factors of 5-10 less than measured experimentally. We believe that this decrease is justified by the fact that of the molecules present in the cell, (1) some may be outside the nucleus, (2) some may be in a phosphorylation state that makes them inactive, and (3) some may be in the process of being produced or degraded. Therefore, we don’t expect that all the molecules present in the cell are competent to function in the spindle.

We have added a discussion of this point to the manuscript in Appendix subsection “Motors and crosslinkers”.

[Editors' note: further revisions were suggested prior to acceptance, as described below.]

The manuscript has been improved but there are some remaining issues that need to be addressed before acceptance.In subsection “Single model perturbations recapitulate the requirement for kinesin-5 motors and CLASP”, the authors added extensive revision on the role of Ase1 crosslinker (and pliable nuclear envelop). Specifically, they stated that in their model, the motor Klp9 can also act as a crosslinker in the absence of Ase1. Therefore, the removal of Ase1 would still produce a bipolar spindle, due to the action of Klp9. Reported data from several labs (McCollum, Millars, Toda, Tran) clearly showed that Klp9 does not function until anaphase. Thus, the authors cannot model the absence of Ase1 as still having crosslinking via Klp9.

Upon further research, we agree that Klp9 is not acting during pre-anaphase and therefore cannot help with bipolar spindle formation. We changed the model so that kinesin-14 has the same force-dependent unbinding characteristics as kinesin-5. In the reference model, the characteristic distance for kinesin14 force-depending unbinding was 3.2 times higher than kinesin-5, making it more likely to unbind. We found that when we reduced the characteristic force to that of kinesin-5 (weaker force-dependent unbinding) spindles were able to form at different lengths and many were able to successfully biorient chromosomes. We updated the figure and the main text with this information.